



# Experimental Study of Regenerative Wind Farms Featuring Enhanced Vertical Energy Entrainment

YuanTso Li[1], Marnix Fijen[1], Brian Dsouza[1], Wei Yu[1], Andrea Sciacchitano[1], and Carlos Ferreira[1]

[1]Delft University of Technology, Faculty of Aerospace Engineering, Kluyverweg 1, 2629 HS Delft, the Netherlands

**Correspondence:** YuanTso Li (Y.Li-18@tudelft.nl)

**Abstract.**

This study presents the experimental validation of regenerative wind farms (RGWFs), a novel wind farm concept designed to enhance overall wind farm performance. RGWFs employ Multi-Rotor Systems with Lifting devices (MRSLs), an innovative wind energy harvesters engineered to promote strong vertical energy entrainment, thereby accelerating wake recovery. In the experiments, MRSLs are scaled for wind tunnel testing, with their rotors modeled using porous disks and their lifting devices represented by airfoils. The tested RGWFs comprise up to three by three MRSLs. Flow quantities within RGWFs and aerodynamic loads on MRSLs are measured using volumetric particle tracking velocimetry and strain gauges, respectively. Compared to conventional wind farms, flow analysis indicates that RGWFs significantly enhance vertical energy entrainment, as evidenced by a more than $200\%$ increase in thrust on second-row-MRSLs. These experimental results, consistent with previous numerical predictions, highlight the promising potential of RGWFs.

## 1 Introduction

In recent decades, offshore wind energy has demonstrated its economic viability as a renewable energy source (Williams and Zhao, 2024). The profitability of offshore wind farms has largely been driven by cost reductions achieved through the close spatial arrangement of turbines and the continual increase in turbine size (Sørensen and Larsen, 2021). However, the levelized cost of energy (LCoE) for offshore wind remains relatively high compared to other technologies, such as gas combined cycle and solar photovoltaic systems (Lazard, 2024). Furthermore, since the most economically favorable sites have already been developed as well as the relative costs of curtailment and energy storage systems rise with increasing total capacity (Wind Europe, 2024; Lazard, 2024), the LCoE of offshore wind could increase further in the foreseeable future. These factors render offshore wind more susceptible to macroeconomic challenges (McCoy et al., 2024). For instance, several high-profile offshore wind projects in North America and Europe have been delayed or canceled (partly) due to escalating interest rates (Empire Wind, 2025; Ørested, 2025; McCoy et al., 2024). In light of these challenges, the current offshore wind market urgently calls for engineering solutions to reduce the LCoE and enhance the industry's competitive position.

One of the primary physical limitations preventing further reductions in the LCoE of offshore wind is the losses caused by turbine–turbine wake interactions within wind farms, which can reduce overall farm efficiency by 10 to $25\%$ (Barthelmie et al., 2009, 2010). Consequently, mitigating wake losses holds significant potential for improving wind farm performance and, in





turn, lowering the LCoE of offshore wind. To address this issue, Ferreira et al. (2024) recently proposed a novel design of wind energy harvesting machine, which is the multi-rotor system with lifting devices (MRSL). MRSL comprises an array of wind turbine rotors mounted on a scaffolding structure, as depicted in Figure 1. In addition, it incorporates large stationary wing elements that generate strong tip vortices. These vortices are intended to enhance mixing between the wake and the ambient

flow, thereby accelerating the wake recovery process. Conceptually, the lifting devices operate similarly to the vortex generators used on aircraft wings, albeit on a much larger scale (Ferreira et al., 2024). Notably, as mentioned by Li et al. (2025c), it is the lifting devices being the essential component to enhancing the wake recovery rate, regardless of the specific implementation approach. In other words, as long as the large-scale lift-generating elements are introduced, the concept of MRSL is expected to be effective. Among possible implementations, the MRSL configuration shown in Figure 1 is considered to be the most

practical realization of this concept by the authors, as the scaffolding structure readily enables the integration of the lifting devices.

Recent studies have shown that the lifting devices of an isolated MRSL can significantly accelerate its wake recovery, both experimentally and numerically. In particular, Broertjes et al. (2024) tested a scaled MRSL in an open-jet wind tunnel. The tested configuration consisted of an array of 16 vertical-axis wind turbines and two wings, with overall dimensions of $1.43$ m

in width and $1.35$ m in height. Their results indicated that the MRSL's wake is effectively deflected vertically due to the influence of the lifting devices. However, due to the spatial limitations of the wind tunnel, their analysis is restricted to the near-wake region ($x/D > 2$, where $x$ is the streamwise distance and $D$ is the width of the tested MRSL). On the numerical side, studies of Avila Correia Martins et al. (2025) and Li et al. (2025b) employ computational fluid dynamics (CFD) solvers with actuator-based methods. Their results show good agreement with the experimental findings of Broertjes et al. (2024)

in the near-wake region and provides evidence that the lifting devices are able to significantly enhance vertical mixing and accelerate wake re-energization in the far wake ($3 < x/D < 8$). Regarding the wind farms composed of MRSLs, which is termed regenerative wind farms (RGWFs) by Ferreira et al. (2024), a recent numerical study by Li et al. (2025c) using CFD with RANS (Reynolds-averaged Navier–Stokes equations) approach showed that wake-induced power losses within RGWFs are reduced to approximately one-third of those observed in wind farms using multi-rotor system without lifting devices. In

other words, the studied RGWFs achieved roughly twice the efficiency of their conventional counterparts.



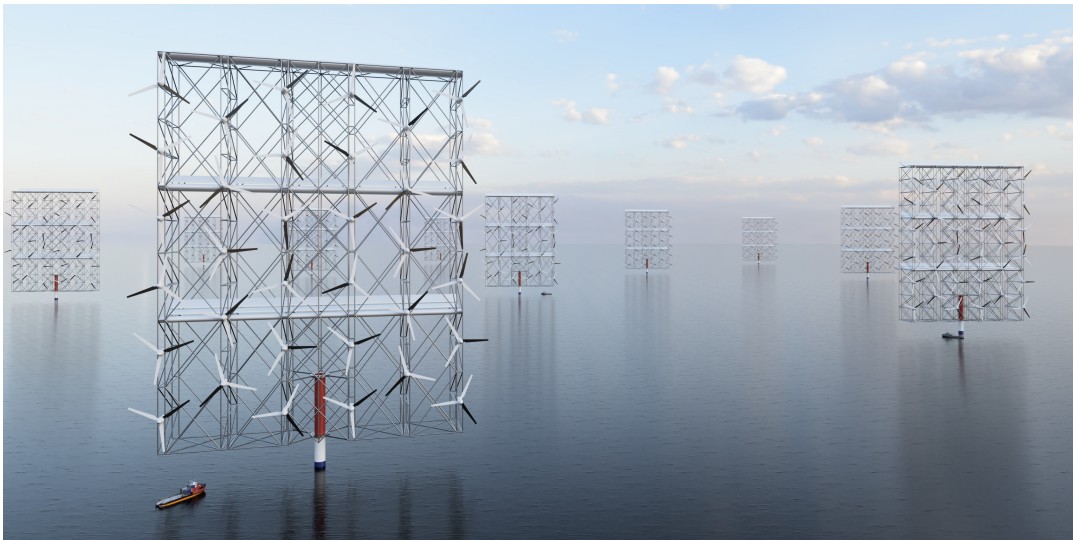

**Figure 1.** Computer-rendered visualization envisioning the multi-rotor system with lifting devices (MRSLs) and a regenerative wind farm (RGFW) deployed in offshore locations. In the current scenario, MRSL is designed to be approximately 300 m in both width and height, with a clearance of 30 m above the sea surface. Courtesy of Carraro et al. (2024).

To further evaluate the practical viability of RGWF, this study conducts comprehensive wind tunnel experiments with a scaled wind farms having an array of $3 \times 3$ or $3 \times 2$ MRSLs. In these experiments, MRSLs are aerodynamically represented by porous disks with wings attached. Three MRSL configurations are tested. The first is designed to eject the wake upward (**Up-Washing**), the second is configured to direct the wake downward (**Down-Washing**), the last is a reference configuration that is without the lifting devices (**Without-Lifting**) for benchmarking. Both the aerodynamic loads on MRSLs and the surrounding flow fields are measured. The flow fields are captured using three-dimensional particle tracking velocimetry (3D-PTV). Furthermore, wind farm layouts with MRSLs aligned and staggered with respect to the inflow direction are both examined.

## 2 Methodology

### 2.1 Wind tunnel and general experimental setup

The scaled RGWF are experimentally tested in the Open Jet Facility (OJF) at the aerodynamic laboratories of Delft University of Technology (TU Delft). OJF is an atmospheric closed-circuit open-jet wind tunnel, and it features an octagonal outlet measuring $2.85 \times 2.85$ m$^2$ (see Figure 2). Additional specifications of OJF are provided in Lignarolo et al. (2014).

Figures 2 and 3 present the CAD (computer-aided design) drawings and a photograph overviewing the general setup of the tested regenerative wind farms. The corresponding CAD files and footage taken during the experimental campaign are available in the associated data repository (Li et al., 2025a). The elements of the setup are elaborated in the following subsections.





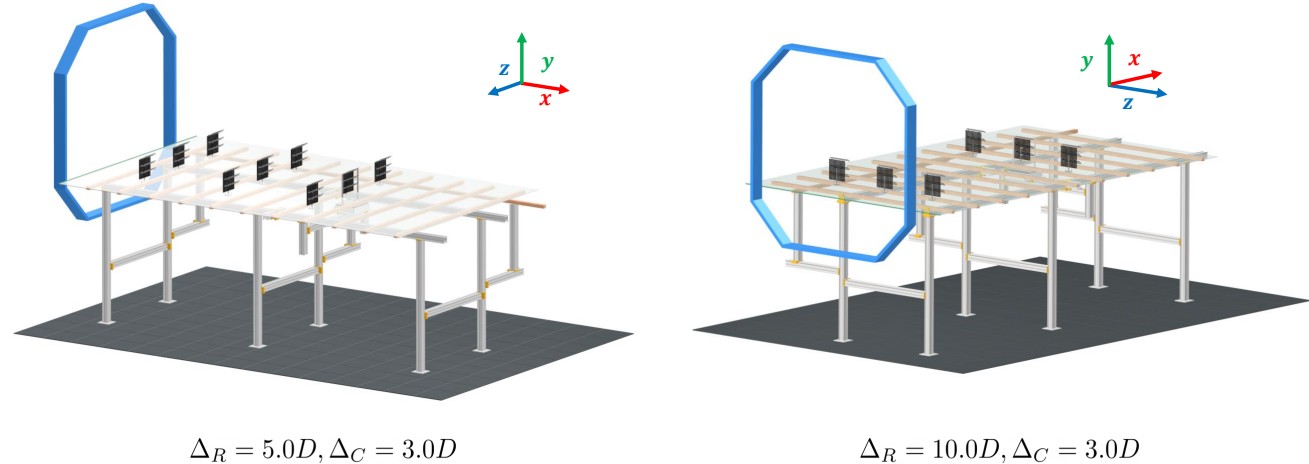

$$\Delta_R = 5.0D, \Delta_C = 3.0D \qquad\qquad \Delta_R = 10.0D, \Delta_C = 3.0D$$

**Figure 2.** Illustration of the experimental setups for the regenerative wind farms with the aligned layout. The left panel shows a wind farm with a row spacing of $\Delta_R = 5.0D$ and MRSLs in the **Up-Washing** configuration (**UW-05**). The right panel shows a wind farm with $\Delta_R = 10.0D$ and MRSLs in the **Down-Washing** configuration (**DW-10**). The blue octagons represent the wind tunnel outlet. Note that the force measurement devices are installed in the last row of each setup.

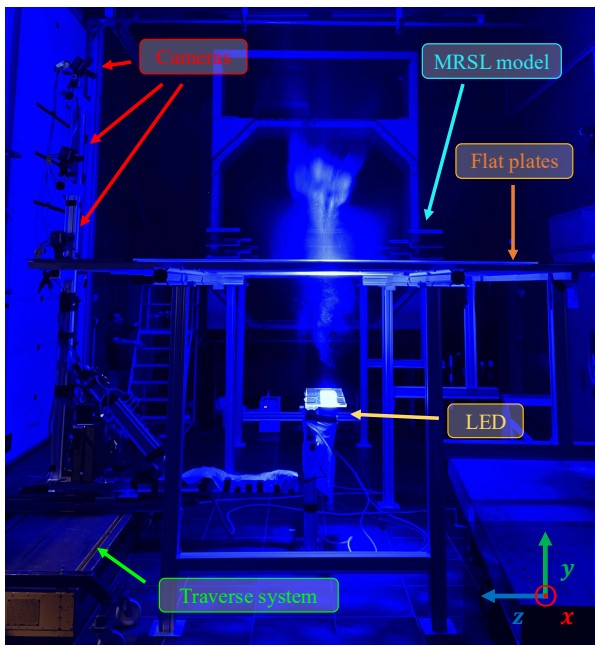

**Figure 3.** Photograph of the experimental setup with key components labeled. The image is taken during flow field measurements for case **DW-05** (see Section 2.7).





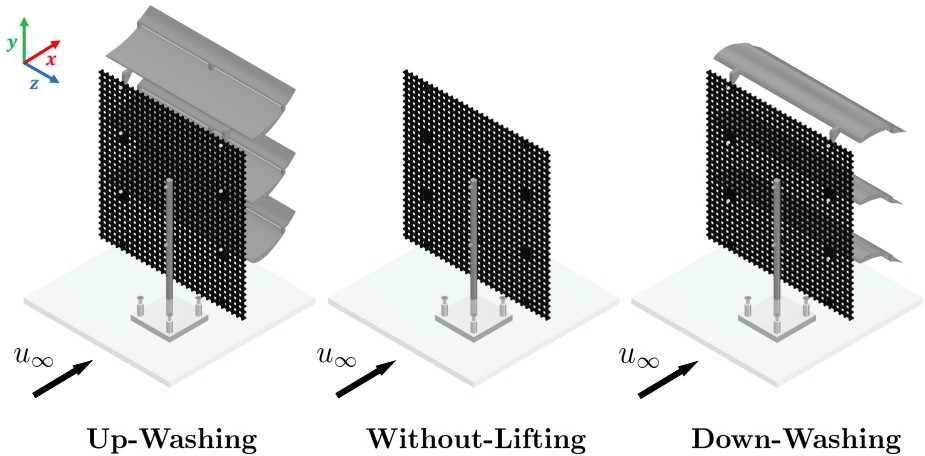

| Up-Washing | Without-Lifting | Down-Washing |

**Figure 4.** Perspective view of the MRSL models used in the experiments. The corresponding configurations are labeled at the bottom. The flow direction is from bottom-left to top-right. CAD models are available in the corresponding data repository (Li et al., 2025a).

## 2.2 Scaled MRSL model

Following the previous studies (Li et al., 2025b, c), ulti-rotor system with lifting devices (MRSLs) are thought to be around 300 m and 300 m in width and height when they are in utility scale. Testing these machines with wind tunnel at this scale is financial prohibitive or simply impossible. Therefore, MRSL are aerodynamically scaled in the current experiments, and these

models are shown in Figure 4. The three different MRSL's configurations, **Up-Washing** (**UW**), **Without-Lifting** (**WL**), and **Down-Washing** (**DW**), are displayed from left to right. These models have a dimension of $300 \times 300$ mm$^2$, making testing them in OJF feasible. Additionally, as illustrated in Figure 4, the rotor components of the MRSL model, i.e. the wind energy harvesting elements, are simplified as a single square porous disk, which is a well-established approach to model the wake effects of wind turbines (Yu, 2018). This simplification are made to avoid the high rotational frequency of small-scale turbine

models, which can exceed several thousand revolution per minute (rpm) for them to maintain the tip speed ratio. Furthermore, the porosity of the disk can be tuned to model a specific thrust coefficients ($C_T$) for wind turbines or MRSLs. In this study, a porosity of $60\%$ is selected, yielding a $C_T$ of 0.72 and giving no reverse flow in the wake. The disk is made of laser-cut acrylic, and its detailed specifications are provided in Figure 5. In the figure, the critical dimensions and the bolting layout for attaching the rod and wings highlighted. Note that the disks are deemed semi-permeable to the flow tracers, as the hole size ($8 \times 8$ mm$^2$)

is significantly larger than the tracer diameters (300 to 400 $\mu$m, see Section 2.5).

The lifting devices of the MRSL in this study consist of three wings. A two-element airfoil profile is selected for the wings due to its relatively high lift coefficient. A detailed rationale for this choice, along with the XY-plot of the airfoil geometry, is provided in Appendix B1. The span of each wing matches the side length of the MRSL, denoted as $D$, and the chord length is set to $c = D/3$. The wings are fabricated with additive manufacturing (3D printing) using PLA (polylactic acid) and are

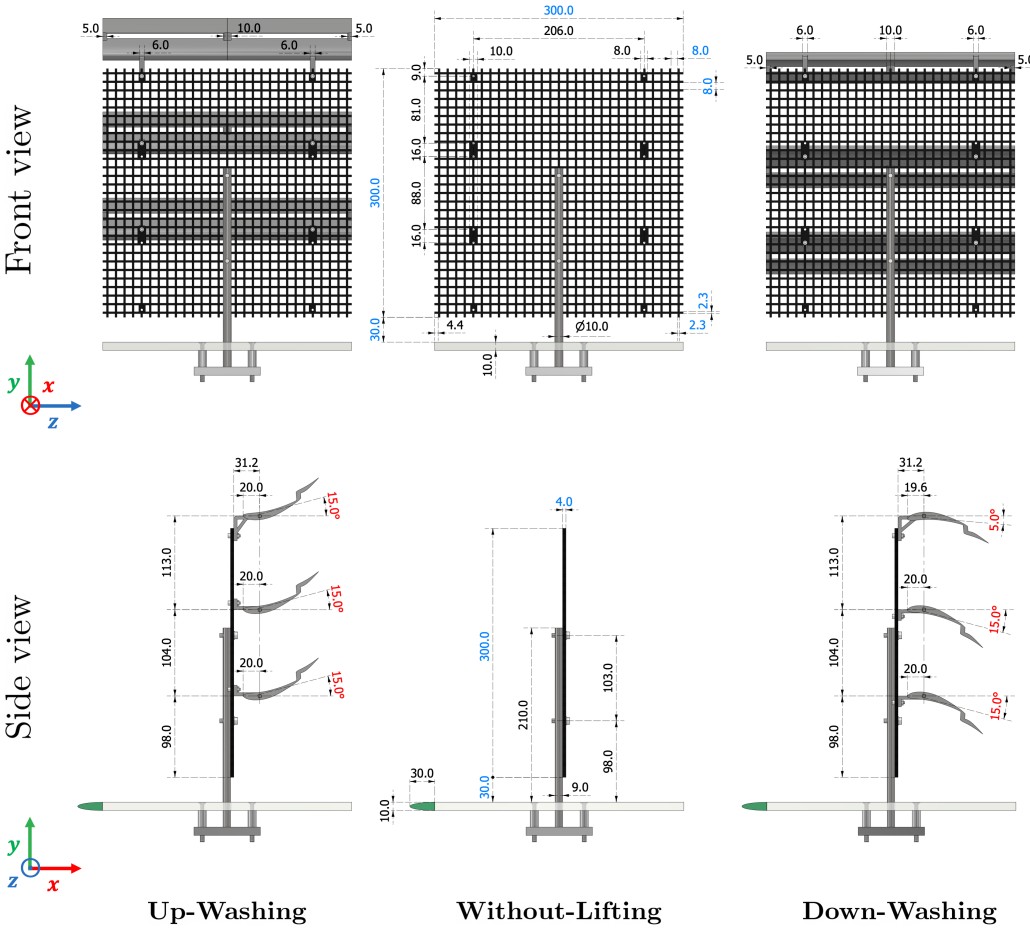

**Figure 5.** Specifications of the MRSL in different configurations. The top and bottom rows show the front and side views, respectively. The left, middle, and right columns correspond to the model representing **Up-Washing**, **Without-Lifting**, and **Down-Washing** configurations, respectively. Critical dimensions are marked in blue, while the pitch angles of the MRSL's wings ($\theta_p$) are shown in red. Other specifications are labeled in black. All lengths are given in millimeters (mm) and angles are in degrees (°). The green semi-ellipses in the side views represent the profile of the leading edge installed upstream of the flat plates.





integrated with supporting structures. These structures include bolting points for attaching the wings to the porous disks, as shown in Figure 5.

The vertical positions of the wings, defined by the locations of their pitching axes (see Figure B3), are at $1.05D$, $0.67D$, and $0.33D$ above from the bottom edge of MRSL. The wings are located downstream of the porous disk, with a streamwise distance of $0.10D$ between the pitching axes and the disk. The clearance between the leading edge of the wings and the disk is $0.04D$

(11 mm). The pitch angles of the wings, $\theta_p$, are determined based on preliminary experimental tests described in Appendix B2. Those tests show that the MRSL configured with these selected pitch angles is capable of generating lift comparable to its thrust. Li et al. (2025c) have shown that MRSLs are effective when their magnitudes of lift approximately matches that of thrust.

The MRSL in this work is held upright by a steel rod having a diameter of 10 mm. The side of the rod in contact with the

porous disk is machined flat to ensure a seamless interface, providing a sturdier installation for both the MRSL (porous disk) and the strain gauges (see Section 2.4). This machining reduces the rod's apparent diameter in side view from 10 mm to 9 mm, as illustrated in Figure 5.

### 2.3   Setups of the regenerative wind farm

The floor, representing the sea surface of the regenerative wind farm, is constructed by seven sheets of transparent flat plates

(plexiglass), each measuring 244 cm, 122 cm, and 1 cm in length, width, and thickness. Assembly details are provided in Figures 6 and 7 for the cases with the aligned wind farm layout and the staggard wind farm layout, respectively. These plates are supported by wooden beams resting on an metallic frame (LINOS X95 System) and are positioned approximately 73 cm above the bottom edge of the wind tunnel exit (7 cm above where the wind tunnel begins to converge in width, see Figure 3). To mitigate flow separation and reduce turbulence generation, an elliptical profile with an aspect ratio of 3 is added to the

leading edge (Hanson et al., 2012). This profile is illustrated in Figure 5 and is fabricated using additive manufacturing. To fasten MRSLs to the flat plates, several holes are drilled and secured using sunken bolts, as shown in Figure 5. Most of the supporting structures for the MRSLs are positioned beneath the flat plates to minimize flow disturbance.

As illustrated in Figure 6, the RGWFs with the aligned layout includes nine available loci for mounting MRSL, arranged in three rows and three columns. The row spacing between these loci is $5D$ (150 cm), and the column spacing of them is $3D$

(90 cm). The distance from the 1$^{st}$-row MRSLs to the leading edge of the flat plates is $2.6D$ (77 cm). Note that not all the aligned cases utilize all nine loci (see Section 2.7).

In terms of the case with staggered wind farm layout, RGWF consists of only two rows of MRSLs, as depicted in Figure 7. The absolute position of its origin is relocated $5D$ downstream relative to the aligned cases (see Figure 6 for reference). This staggered layout is achieved by shifting three of the flat plates of the aligned layout by a distance of $D$ in the lateral direction.

Further detail dimensions of the RGWF's layouts, including the relative positions of the flat plates to the wind tunnel, the locations of the fields of view (FOVs), and other relevant setup details, are also labeled in Figures 6 and 7. The origin of RGWF is defined at the top of the porous disk of the 1$^{st}$-row-center-column MRSL (marked as "$\otimes$" in Figures 6 and 7), and the $x$,





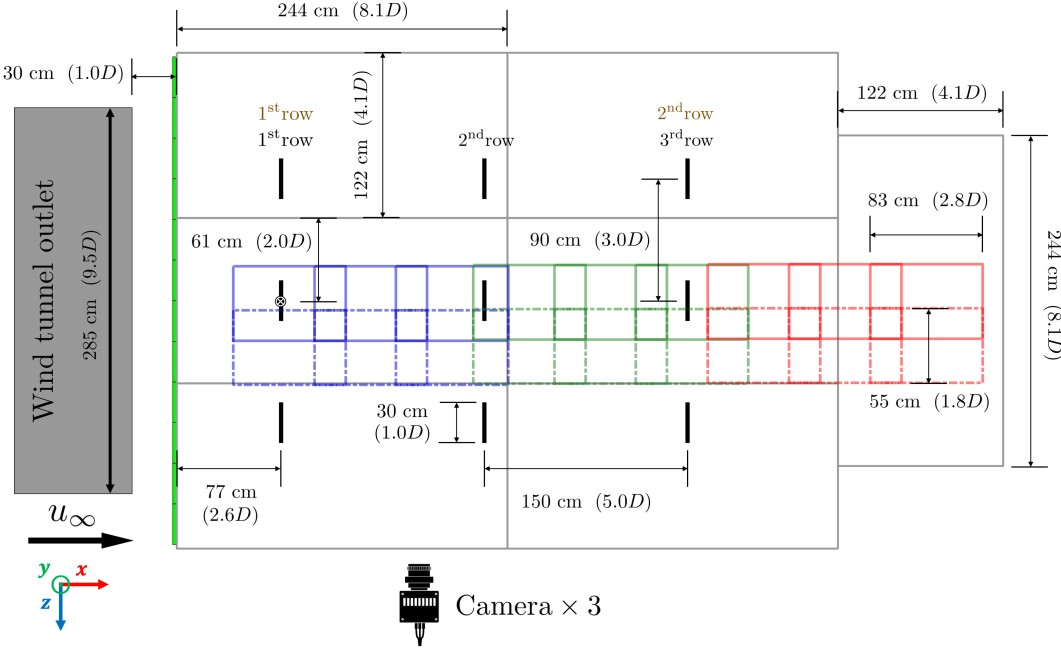

**Figure 6.** Layout of the experimental setups for the regenerative wind farms, with dimensions labeled. The viewing angle is from above, with the wind being blown from left to right. Seven plexiglass flat plates, representing the sea surface, are shown as gray rectangles, and the MRSL loci are marked with thick black lines. The origin of the coordinate system, located at the positions of 1st-row-center-column MRSL, is marked with "⊗". Row numbers are shown in black for cases with $\Delta_R = 5D$ and in brown for those with $\Delta_R = 10D$. The field of views of the 3D-PTV are represented by blue, green, and red rectangles, which the colors correspond to the three locations for the traverse system (see Section 2.8). Green elongated stripes indicate the elliptical leading edges placed in front of the flat plates, and the wind tunnel outlet is shown as a gray block. During the experiment, the cameras are oriented to shoot from the positive $z$-direction.

$y$, and $z$ directions correspond to the streamwise, vertical, and lateral directions, respectively. The positions of the other key components, such as the cameras, light sources, and the traverse system, are shown by a photograph in Figure 3.

## 2.4 Force measurement methodology

To quantify the forces exerted by MRSLs, both streamwise (thrust and drag) and vertical (lift) forces are measured for the MRSLs located in the mid-column. Streamwise forces are recorded using two one-component strain gauges (KD24s 10N, ME-Me$\beta$systeme) which are capable of measuring tensile and compressive loads with a precision of $0.1$ N. Vertical forces are measured using a bench scale (FKB 6K0.02, KERN) with a precision of $4 \times 10^{-4}$ N. The measurement duration is about 30 s
for both forces. Sampling frequencies are around $1,000$ Hz for the streamwise forces and $1$ Hz for the vertical force.

Since the aerodynamic forces on the disk and wings are coupled, measurements are conducted both with the wings attached and detached, as illustrated in Figures 8. Notably, the wing holders shown on the right side of Figure 8 do not contact either the porous disk or the flat plates. Additional holes are drilled on the flat plates for this purpose and are taped when not in use.





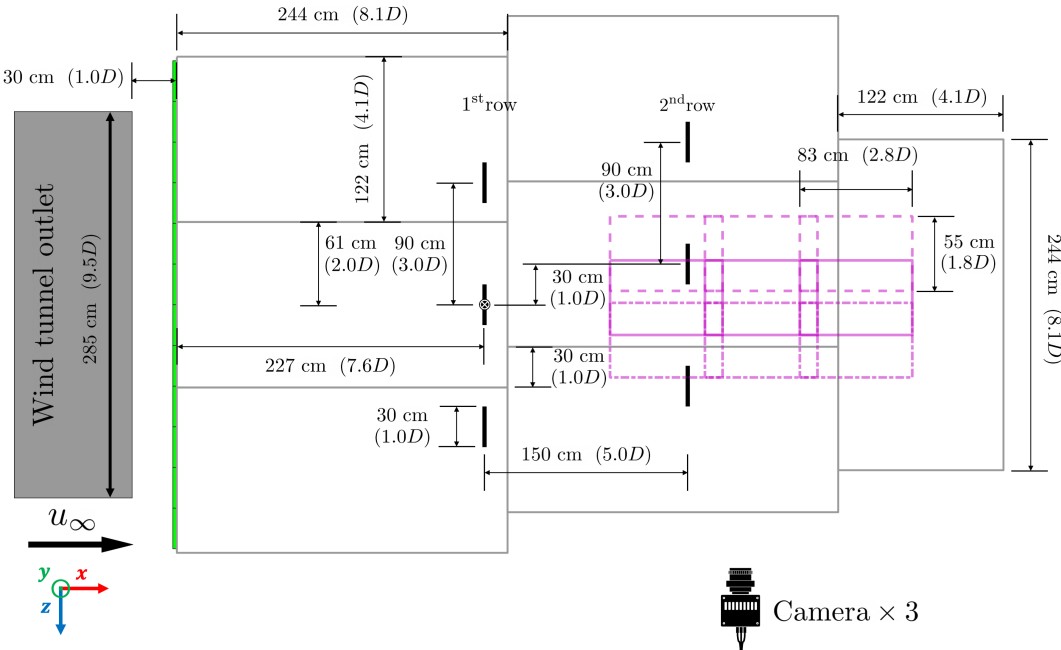

**Figure 7.** Layout of the experimental setups for regenerative wind farms with dimensions labeled for the staggered case (case **UW-05-ST** in Table 1). The viewing angle is from above, with the wind being blown from left to right. The origin of the coordinate system, located at the positions of $1^{st}$-row-center-column MRSL, is marked with "⊗". The loci for placing MRSLs are indicated by thick black lines. Magenta rectangles with solid, dashed, and dotted-dashed lines depict the field of views of the 3D-PTV system.

By subtracting the streamwise forces measured in the two setups, the thrust from the disk and the drag from the wings can be
determined separately, despite their aerodynamic coupling. The integration of force measurement devices into the wind farm setup is also shown in Figure 2. Note that these force measurement devices are removed when measuring the flow fields.

## 2.5 Flow measurement methodology

This study employs three-dimensional particle tracking velocimetry (3D-PTV) to measure flow quantities based on particle trajectories obtained through imaging system. The PTV system consists of three high speed cameras (Photron Mini AX100)
for imaging and arrays of light-emitting diodes (LED-Flashlight 300 blue, LaVision GmbH) for illumination. Both the cameras and the light source are mounted on a traverse system as shown in Figure 3. This allows the PTV system to scan through RGWF with ease. Neutrally buoyant helium-filled soap bubbles (HFSB) are used as flow tracers. The flow traces are dispensed by an in-house developed seeding system, and their median diameter is approximately 300 to 400 $\mu$m (Scarano et al., 2015; Faleiros et al., 2019). With the current setup, the field of view (FOV) for a single volume is approximately 830 mm, 550 mm,
and 880mm, corresponding to $2.8D$, $1.8D$, and $2.9D$ in the streamwise, lateral, and vertical directions, respectively. This set





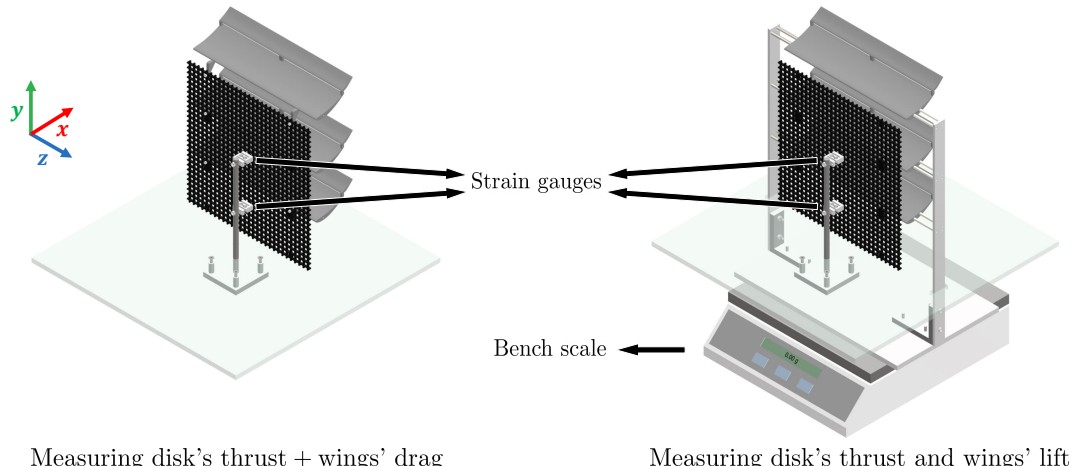

Measuring disk's thrust + wings' drag          Measuring disk's thrust and wings' lift

**Figure 8.** Setups for measuring the forces exerted by the MRSL. Left: Configuration for measuring the total streamwise force, comprising the disk's thrust ($T^R$) and the wings' drag ($D^W$). Right: Configuration for concurrently measuring the disk's thrust ($T^R$) and the wings' lift ($L^W$). Note that the wings' drag ($D^W$) is not captured in the right-hand setup, as the wings are detached from the porous disk.

up gives a resolution of 0.7 mm/px. For each measurement, $3,000$ images are captured with the cameras operating at $1,000$ Hz. Further detail about the 3D-PTV specifications and setup are provided in Appendix A.

## 2.6 Image processing for flow field quantities

The general procedure of image-processing to quantify the flow fields is outlined as follow. First, a subtract-average time filter
with a window of 11 timesteps is applied to the raw images to reduce noise. Next, particle tracks are extracted from the filtered images using the Shake-the-Box (STB) algorithm (Schanz et al., 2016). The resulting tracks are then binned onto a Cartesian grid to convert the Lagrangian data into Eulerian fields. Finally, the binned Eulerian fields from each measurement volume are stitched together to construct a complete flow field of RGWF. Detailed descriptions of each step are provided in the remainder of this subsection.

### 2.6.1 Particle tracking

The images are processed using the Shake-the-Box (STB) algorithm (Schanz et al., 2016) to reconstruct the particle tracks. Three passes are used when performing STB algorithm, which are forward in time, backward in time, and reconnection of interrupted tracks. With this setup, the number of the active tracks per volume at an instant ranges from 4 k to 20 k. The variability in track count is primarily due to the blockage of MRSLs, including optical obstructions and blockage of particle
seedings.

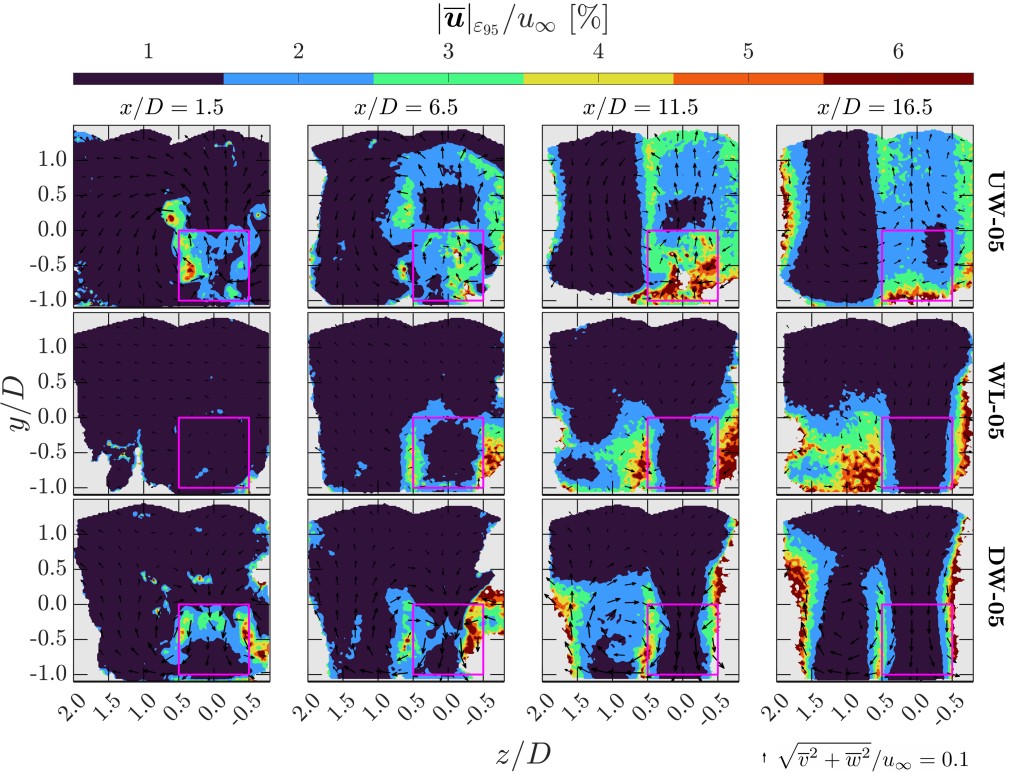

**Figure 9.** Contours of the uncertainty in the time-averaged velocity magnitude ($|\overline{\boldsymbol{u}}|_{\varepsilon_{95}}$) for cases with a row spacing of $5D$ at several $x$-planes. Case labels given in Table 1 are indicated on the right side of each row, and the corresponding $x$-positions are labeled at the top of each column. The projected areas of MRSLs are marked with magenta squares. Arrows indicate the direction of in-plane velocities, with their lengths scaled by the in-plane velocity magnitude. The scaling reference for the arrows is shown at the bottom right of the figure.

### 2.6.2 Binning and uncertainty

Once the particle tracks are obtained, they are binned onto a Cartesian grid to compute the spatiotemporal statistics of the flow fields. The binning window consists of $16 \times 16 \times 16$ cubic voxels ($11.2 \times 11.2 \times 11.2$ mm$^3$) with $50\%$ overlap. Spatial fitting is performed using a second-order polynomial as advised by Agüera et al. (2016). Tracks shorter than 10 timesteps are excluded
from the analysis, and no time filtering is applied.

In general, the particle count $N$ per bin exceeds $1,000$. However, in regions near the MRSL models and in swirling flow areas, the particle count may drop to 300 or fewer, as the lines of sight emanating from the cameras can be obstructed by the MRSL structures. The expanded uncertainty at $95\%$ confidence level of the mean velocity, denoted as $|\overline{\boldsymbol{u}}|_{\varepsilon_{95}}$, is defined in Equation (1) (Sciacchitano and Wieneke, 2016). As shown in Figure 9, in regions that are not significantly influenced by the
MRSL, $|\overline{\boldsymbol{u}}|_{\varepsilon_{95}}/u_\infty$ is typically well below $1\%$. In contrast, in the wake regions, the uncertainty is substantially higher, primarily due to the reduced particle count $N$. This effect is particularly pronounced in cases with the without-lifting configuration,



where tracers are partially blocked by the porous disks and the replenishment is limited due to weaker mixing. Nevertheless, $|\overline{\boldsymbol{u}}|_{\varepsilon_{95}}/u_\infty$ generally remains below $5\%$, indicating the obtained statistics have converged.

$$|\overline{\boldsymbol{u}}|_{\varepsilon_{95}} \stackrel{\triangle}{=} Z_{95} \times \frac{\sqrt{\sigma_u^2 + \sigma_v^2 + \sigma_w^2}}{\sqrt{N}}, \qquad Z_{95} = 1.96 \tag{1}$$

### 2.6.3 Stitching and scaling the flow quantities

Since the region of interest for RGWF is significantly larger than the 3D-PTV system's FOV, multiple measurement volumes are acquired to cover the entire domain. These volumes are subsequently stitched together to reconstruct the complete flow fields of RGWFs. The stitching process uses a hyperbolic tangent function as a weighting function to smoothly blend the overlapping regions. Additionally, small variations in wind tunnel speed across different volumes are compensated by scaling the flow quantities during the stitching process. A detailed description of the stitching methodology is provided in Appendix C.

After stitching the measured volumes, the total flow field domain for cases with the aligned layout spans $-1.3 < x/D < 16.8$ (streamwise), $-0.8 < z/D < 1.9$ (lateral), and $-1.1 < y/D < 1.5$ (vertical) relative to the defined origin (see Figure 6). While for the case with the staggered layout, the total flow field domain spans $3.1 < x/D < 10.5$, $-2.7 < z/D < 2.2$, and $-1.1 < y/D < 1.5$ (see Figure 7).

Finally, to account for the slight velocity differences across experimental cases caused by wind tunnel variability, the stitched flow fields are normalized by a reference volume-averaged $\overline{u}$, which serves as the inflow wind speed $u_\infty$. For aligned cases, the reference volume is defined at $-0.95 < x/D < -0.60$, $0.53 < y/D < 0.88$, and $0.84 < z/D < 1.20$, while the reference volume for the staggered cases is $3.77 < x/D < 4.14$, $-1.50 < z/D < -1.22$, and $0.90 < y/D < 1.28$. For the current measurement, the obtained $u_\infty$ is around 7.3 m/s.

### 2.7 Test Matrix

Seven RGWF configurations are tested in this study, and they are listed in Table 1. In this study, RGWFs with three different MRSL configurations are tested with the aligned layout (see Figure 6), which are **Up-Washing** (**UW**), **Without-Lifting** (**WL**), and **Down-Washing** (**DW**). To better understand the development of MRSL's wakes and to investigate the effect of row spacing $\Delta_R$ on MRSL's performance, cases with $\Delta_R = 5D$ and $10D$ are examined for all three configurations with the aligned wind farm layout (see Figure 2). Moreover, an additional case with a staggard layout is tested (see Figure 7), which is case **UW-05-ST** in Table 1. This cases is tested to preliminarily explore the impacts of apparent wind farm layout, such as those caused by varying incoming wind directions in real-world scenarios.

### 2.8 Procedure of the experiments

This part summarizes the procedures followed during the experimental campaign. Footage of the experiments demonstrating how the procedure is executed is available in the accompanying data repository (Li et al., 2025a).



**Table 1.** Regenerative wind farms tested in this study. The first part of each case label denotes the MRSL configuration, where **UW**, **WL**, and **DW** stand for **Up-Washing**, **Without-Lifting**, and **Down-Washing**, respectively. The two digits in the end/middle indicate the row spacing $\Delta_R$, with **05** and **10** corresponding to $5D$ and $10D$, respectively. The suffix **ST** stands for staggered, indicating that MRSLs in consecutive rows are not aligned with the inflow direction. For all RGWF configurations, the number of columns is fixed at three, and the lateral spacing between adjacent columns is $3D$.

| Case label | MRSLs' configuration | $\Delta_R$ | Number of rows | Load measurement |
|---|---|---|---|---|
| **UW-05** | **UW** | $5D$ | three | yes |
| **WL-05** | **WL** | $5D$ | three | yes |
| **DW-05** | **DW** | $5D$ | three | yes |
| **UW-10** | **UW** | $10D$ | two | yes |
| **WL-10** | **WL** | $10D$ | two | yes |
| **DW-10** | **DW** | $10D$ | two | yes |
| **UW-05-ST** | **UW** | $5D$ | two | no |

When measuring the flow fields of the aligned cases, the traverse system where PTV system is mounted on is initially positioned at the most upstream location, allowing the access to the FOVs labeled in blue in Figure 6. For each RGWF configuration, all FOVs at this traverse position are scanned before reconfiguring RGWF. This re-configuration involves removing and re-installing MRSLs. After completing all the six cases with the aligned wind farm layout in this traverse location, the traverse system is manually moved to the next downstream location, and the same procedure is repeated. This approach minimizes the need of re-positioning and re-calibrating the PTV system, which is relatively more time-consuming. In total, three traverse positions are used for the cases with the aligned wind farm layout, requiring two manual re-positions of the traverse system. These three traverse locations correspond to the blue, green, and red FOVs shown in Figure 6. After completing the flow measurement of all the aligned cases, the case with staggered wind farm layout is tested. For the staggered case, the traverse system remains fixed throughout the measurements, covering the nine magenta FOVs in Figure 7.

In terms of load measurements, the force measuring devices are always positioned at the locus in the mid-column of the last row (see Section 2.9). MRSLs in the other rows are installed or removed depending on the specific measurement objective. For example, to characterize the forces exerted by an MRSL in the 1st row, the MRSL of interest is installed at the 3rd-row-locus (where the sensors are located), while the MRSLs at the 1st- and 2nd-row-loci are removed. This approach is considered appropriate because the flow field does not exhibit significant variation in the streamwise direction until $x/D > 12.0$, as shown in Figure 10. Additionally, since the smallest row spacing used in this study is $\Delta_R = 5D$, the influence of downstream MRSLs on the flow conditions experienced by upstream MRSLs is expected to be negligible.

Regarding the duration of the experimental campaign, flow field measurements span approximately 1.5 weeks, while the load measurements are completed within a single day. Note that due to the scheduling constraint of the wind tunnel, only the up-washing configuration is tested for the staggered layout. Also, the force measurements are not conducted for the staggered case due to the same reason.





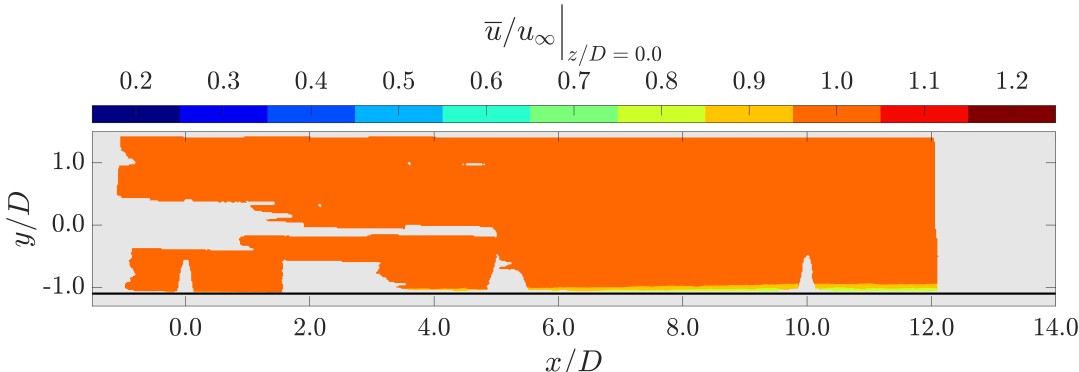

**Figure 10.** Contours of time-averaged streamwise velocity $\overline{u}$ when all MRSLs are absent. The value for $u_\infty$ is 7.3 m/s. The position of the origin and the coordinate system are given in Figure 6.

## 2.9 Flow field characteristics of an empty reference case

To characterize the flow field in the region of interest when all MRSLs are absent, the time-averaged streamwise velocity ($\overline{u}$) is measured when all MRSLs are dis-mounted and is presented in Figure 10. Note that this reference case generally follows the recipe and the procedure introduced earlier. However, to reduce the acquisition time, only $1,500$ images instead of $3,000$ are taken per each volume, and the measurements end at $x/D = 12$.

The velocity contour in Figure 10 shows that the incoming flow while all MRSLs are absent is generally spatially uniform, with minimal vertical velocity shear. Up to at least $x/D = 12.0$, noticeable reductions in $\overline{u}$ are mostly confined to regions below $y/D = -1.0$. This indicates that the RGWF's inflow can still be considered as uniform despite the presence of the flat plates, as the growth of the boundary layer is limited. Note that the large areas where data are not available are due to the lack of tracer particles. This limitation is largely alleviated when MRSLs are introduced as the flow tracers are better dispersed by them.

## 3 Results and discussions

This section presents and analyzes the experimental results. Section 3.1 focuses on the load measurements. Sections 3.2 to 3.4 examine the flow fields through contour plots. Section 3.5 investigates the integral wake characteristics using area-averaged available power. Across all these results, RGWFs are shown to substantially outperform conventional counterparts in both wake recovery rate and overall wind farm efficiency. Additionally, Section 3.6 provides a detailed analysis of the wake recovery mechanisms, offering evidence that RGWFs achieve stronger vertical energy entrainment. This section also compares the present experimental findings with prior numerical studies, enabling cross-validation. Finally, Section 3.7 briefly explores the scenario in which MRSLs in adjacent rows are staggered with respect to the inflow direction. The results demonstrate that the proposed concept remains effective, although with slightly diminished performance.



## 3.1 Forces exerts by MRSLs

This subsection presents the forces exerted by the MRSLs. The measurement methodology and procedure are described in Sections 2.4 and 2.8, and all measurements are conducted with the MRSLs located in the mid-column. In this work, $T^R$ denotes the thrust (streamwise force) exerted by the porous disk (rotor), $L^W$ denotes the lift (vertical force) exerted by the wings, and $D^W$ represents the drag (streamwise force) exerted by the wings. For consistency, the positive directions for $T^R$, $L^W$, and $D^W$ are defined as negative $x$, positive $y$, and negative $x$, respectively.

The normalized forces, denoted as $\widehat{T}^R$, $\widehat{L}^W$, and $\widehat{D}^W$, are defined in Equation (2), where the hat operator ( $\widehat{\ }$ ) indicates normalization. Normalization is performed by dividing the measured forces by the reference thrust $T^R$ measured at the mid-column of the 1st-row MRSL in case **WL-05**, denoted as $T^R\big|_{1^{st}}^{\textbf{WL-05}}$. Unless otherwise specified, the value of $T^R\big|_{1^{st}}^{\textbf{WL-05}}$ used in this study is 2.07 N, corresponding to a thrust coefficient $C_T$ of 0.72. The definition of $C_T$ employed in this work is provided in Equation (3), with $u_\infty = 7.3$ m/s and $\rho = 1.225$ kg/m$^3$.

$$\widehat{T}^R \triangleq \frac{T^R}{T^R\big|_{1^{st}}^{\textbf{WL-05}}}, \qquad \widehat{L}^W \triangleq \frac{L^W}{T^R\big|_{1^{st}}^{\textbf{WL-05}}}, \qquad \widehat{D}^W \triangleq \frac{D^W}{T^R\big|_{1^{st}}^{\textbf{WL-05}}} \tag{2}$$

$$C_T \triangleq \frac{T^R}{0.5\rho\, u_\infty^2 D^2} \tag{3}$$

### 3.1.1 Cases with row spacing being $5D$

The results for $\widehat{T}^R$, $\widehat{L}^W$, and $\widehat{D}^W$ for the cases with a row spacing of $\Delta_R = 5D$ are presented in Figure 11. It is found that $T^R$ measured at the 2nd and 3rd rows are approximately tripled in cases **UW-05** and **DW-05** compared to case **WL-05**, despite the presence of additional drag ($D^W$). These increases are even more pronounced than the numerical results reported by Li et al. (2025c), who found that $T^R$ downstream of the 2nd row approximately doubled with the inclusion of lifting devices. It is important to note, however, that the experiments in this study are conducted under near-laminar inflow conditions with a uniform velocity profile, while the simulations in Li et al. (2025c) applied an inflow turbulence intensity of $8\%$ and a vertical velocity shear.

According to classic actuator disk theory (Manwell et al., 2010), tripling the thrust force $T^R$ implies that the power extracted by MRSLs increases by more than a factor of five. This is based on the relations that $T^R \propto u_\infty^2$ and $P^R \propto u_\infty^3$, where $P^R$ represents the power harvested by the MRSL. With the current results, the energy extracted by the MRSLs positioned in the 2nd and 3rd rows are increased around $400\%$ through the integration of the lifting devices. This substantial enhancement highlights that RGWFs could be much more land-efficient than the conventional wind farms. That is, RGWFs are able to output significantly more power with a given unit sea surface.

In addition to the substantial gains in $T^R$, several aerodynamic characteristics of MRSLs and RGWFs are noteworthy. As shown in Figure 11, the ratio of lift to thrust ($L^W/T^R$) is approximately $100\%$ in the 1st row for cases **UW-05** and **DW-05**, indicating that the MRSLs perform as intended (see Section 2.2). However, this ratio gradually decreases in the 2nd and 3rd




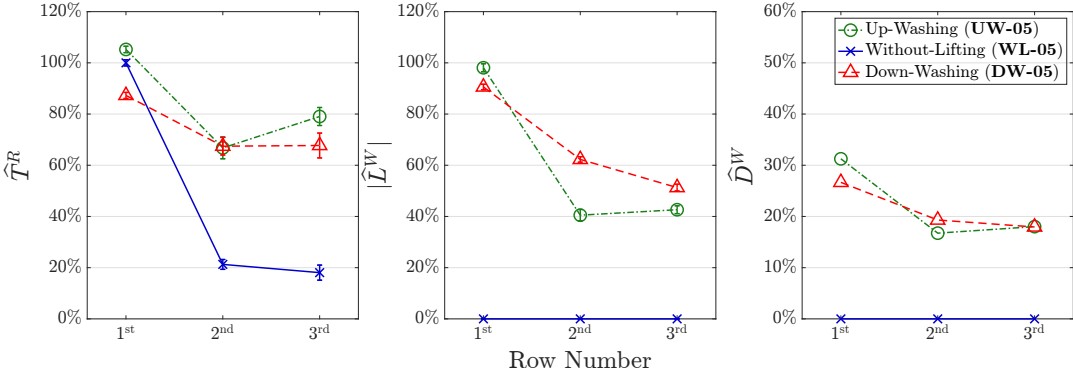

**Figure 11.** Normalized (time-averaged) rotor's thrust ($\widehat{T}^R$), wings' lift ($\widehat{L}^W$), and wings' drag ($\widehat{D}^W$) for cases with row spacing being $5D$. The measured standard deviations for $\widehat{T}^R$ and $\widehat{L}^W$ are labeled with the error bars. The case labels corresponding to Table 1 are indicated in the legends.

rows. This reduction can be attributed to the vertical flow induced by upstream MRSLs, which modifies the inflow conditions for those of the downstream ones. Specifically, the induced vertical velocity alters the effective angle of attack $\alpha$ experienced by the MRSL's wings, while the pitch angle $\theta_p$ of each wing remains fixed regardless of its position in the RGWF (see Section 2.2).

270 This observation suggests that to maintain consistent aerodynamic performance, the MRSL's wings should either incorporate adjustable pitch mechanisms or be designed to remain effective across a range of $\alpha$. Furthermore, in case **UW-05**, the values of $T^R$, $L^W$, and $D^W$ in the 3rd row exceed those in the 2nd row, indicating that wake recovery effects accumulate as the flow progresses deeper into RGWFs. A similar trend is reported in the simulations of Li et al. (2025c). Finally, the error bars in Figure 11 show that the fluctuations (standard deviations) of $T^R$ increase in the downstream rows, reflecting the growing

275 influences of turbulence generated by the MRSLs located upstream.

Another notable observation is that the the 1st-row MRSL of the up-washing configuration exhibits the highest $T^R$ among the three configurations, followed by without-lifting, and then down-washing. This result can be attributed to the bound circulation system generated by the top wings of the MRSL. According to classic airfoil theory, the flow over the suction side of an airfoil accelerates, while the flow over the pressure side decelerates (Anderson, 2011). As a result, in the up-washing configuration,

280 the upper region of the MRSL experiences increased flow velocity, enhancing the thrust generated. Conversely, in the down-washing configuration, the deceleration caused by the pressure side reduces the flow velocity through the upper part of the MRSL, resulting in lower $T^R$. This effect has also been observed in prior numerical simulations by Li et al. (2025b) and Li et al. (2025c), and is further explored in Appendix B2.

### 3.1.2 Cases with row spacing being $10D$

285 To investigate whether the effects of lifting devices are still significant in regenerative wind farms with larger row spacing, cases with $\Delta_R = 10D$ are tested, namely cases **UW-10**, **WL-10**, and **DW-10** in Table 1. The results of the forces exerted in these cases are presented in Figure 12. It is evident that the cases with lifting devices (**UW-10** and **DW-10**) continue to





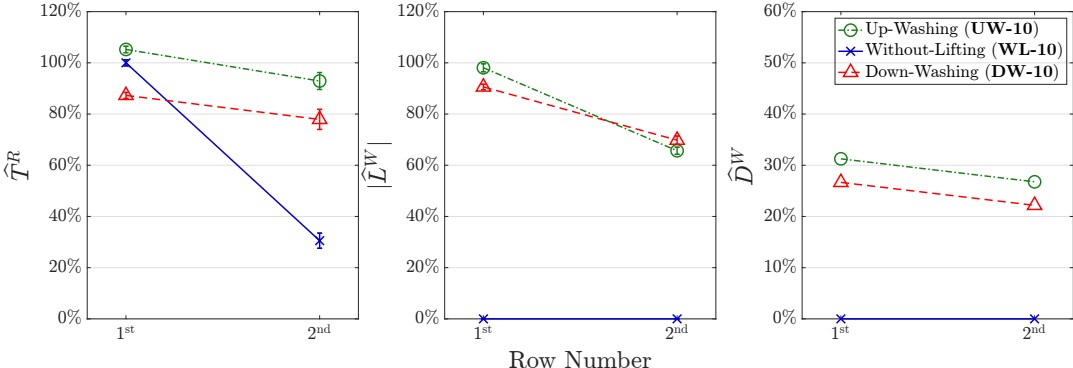

**Figure 12.** Normalized (time-averaged) rotor's thrust ($\widehat{T}^R$), wings' lift ($\widehat{L}^W$), and wings' drag ($\widehat{D}^W$) for cases with row spacing being $10D$. The measured standard deviations for $\widehat{T}^R$ and $\widehat{L}^W$ are labeled with the error bars. The case labels corresponding to Table 1 are indicated in the legends.

substantially outperform the case without lifting devices (**WL-10**), demonstrating the proposed concept remains effective with larger $\Delta_R$. Moreover, the 2nd-row MRSLs in the cases with $\Delta_R = 10$ exert stronger forces compared to the 2nd-row MRSLs with $\Delta_R = 5D$ across all three configurations. These results suggest that larger row spacing allows more wake recovery, which is an outcome consistent with the field measurements (Barthelmie et al., 2010).

## 3.2 Time-averaged streamwise velocity

Contours of the time-averaged streamwise velocity $\overline{u}$ are presented in Figures 13 and 14, where Figure 13 corresponds to cases with $\Delta_R = 5D$ and Figure 14 corresponds to cases with $\Delta_R = 10D$. In addition to the plane $z/D = 0$, several $x$-planes are included to provide a holistic three-dimensional view of the flow fields. The direction and relative magnitude of the in-plane velocities are qualitatively illustrated using arrows. Additionally, to provide more precise illustrations, planar views of the $\overline{u}$ fields at various $x$-planes for cases with $\Delta_R = 5D$ are provided in Figure 15. For a more comprehensive set of velocity contours, including those for the time-averaged vertical velocity $\overline{v}$, readers are referred to the work of Fijen (2025).

Starting with the inflow conditions at the $x/D = -1.0$ plane, all three cases exhibit uniform inflow. Additionally, a deceleration in the streamwise velocity is observed directly upstream of the 1st-row MRSLs, caused by the induction effect of the porous disks. This behavior is consistent with previous experimental findings (Lignarolo et al., 2016), further supporting the validity of the measured velocity fields.

Downstream of the 1st-row MRSLs, regions of velocity deficit, commonly referred to as wakes, are clearly observed. A comparison of the contours in Figures 13 and 15 reveals that the three MRSL configurations produce distinctly different wake patterns, indicating that the lifting devices significantly influence wake aerodynamics. Specifically, in the up-washing configuration, the wake is effectively deflected upward as it progresses downstream, while in the down-washing configuration, the wake is steered downward and subsequently spread laterally. These trends persist throughout the measured domain. Consequently, in both **UW-05** and **DW-05**, the regions with the strongest velocity deficit are diverted away from the frontal areas of

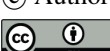

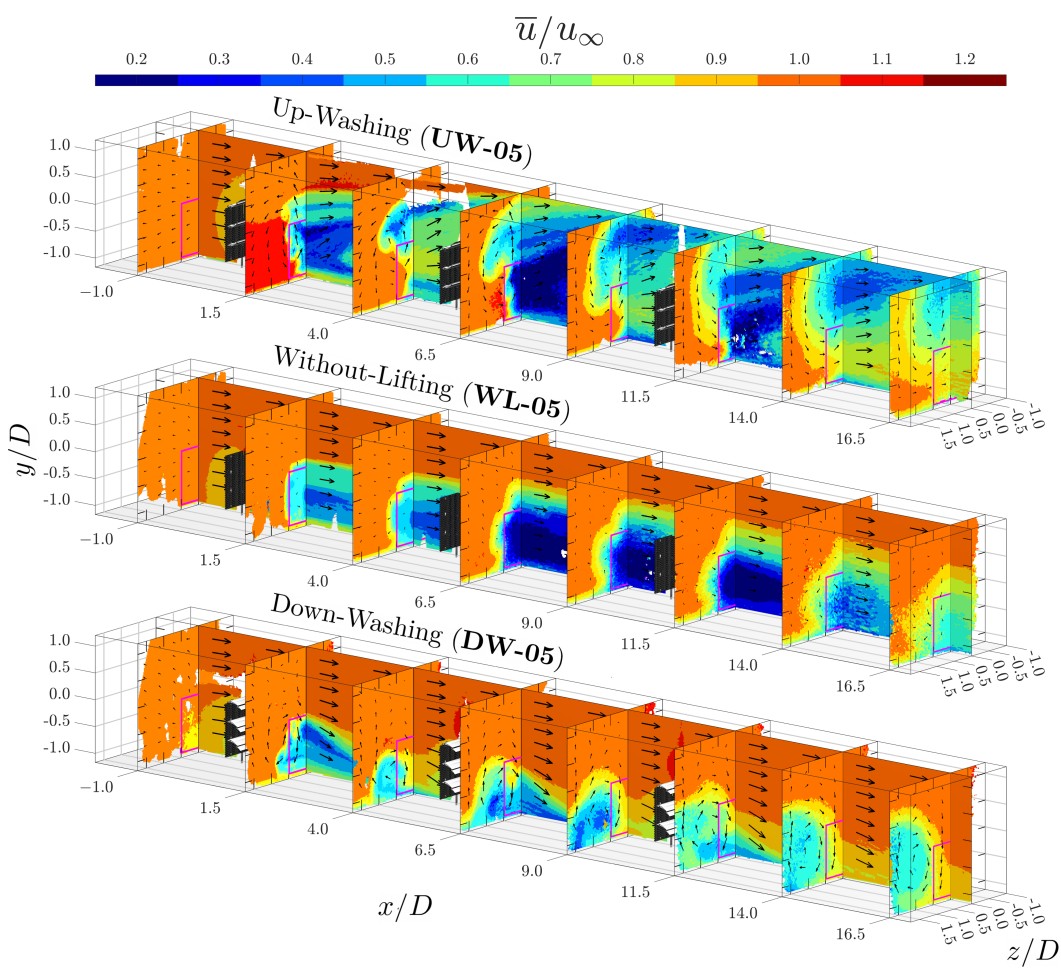

**Figure 13.** Holistic view of the time-averaged streamwise velocity ($\overline{u}$) fields for cases with a row spacing of $5D$, where MRSLs are positioned at $x/D = 0.0$, 5.0, and 10.0. The projected areas of the MRSLs are marked with magenta squares. Arrows indicate the direction of in-plane velocities, with their lengths scaled according to the in-plane velocity magnitudes.





the 2$^{nd}$- and 3$^{rd}$-row MRSLs, enabling those rows to enjoy higher-energized flows that have weaker velocity deficits. In con-
trast, in **WL-05** (without lifting devices), the wake shows minimal vertical or lateral deflection as it is convected downstream,
causing the downstream MRSLs to remain fully immersed in the wake of the upstream ones. (The regions with particularly
strong velocity deficits observed at $x/D = 1.5$ for **WL-05** in Figure 15 are associated with the wake of the support tower and
wing mounting structures.) These flow field observations correlate closely with the force measurements presented in Figure 11,
explaining why the downstream MRSLs in **UW-05** and **DW-05** significantly outperform those in **WL-05**.

A closer examination of the in-plane velocities, as depicted by the arrows in Figures 13 to 15, reveals the presence of
strong swirling motions in the cases where MRSLs are equipped with lifting devices. In contrast, these swirling structures
are absent in the case without lifting devices. It is these swirls that divert the wakes away from the downstream MRSLs.
However, steering the wakes alone is not the ultimate goal of RGWF. The primary objective is to enhance vertical entrainment
by promoting mixing between the wake and the ambient flow above. Based on the presented contours, this goal appears to
be successfully achieved. Specifically, the cases with lifting devices exhibit broader regions where $u/u_\infty < 0.95$ and reduced
areas where $u/u_\infty < 0.15$, indicating increased wake penetration into the surrounding flow and that the most severe velocity
deficits are mitigated. These two indicators collectively demonstrate that the lifting devices effectively enhance vertical mixing
and promote wake recovery.

Figure 14 presents the streamwise velocity contours for the three cases with a row spacing of $\Delta_R = 10D$. In general, these
cases exhibit similar flow characteristics to those with $\Delta_R = 5D$ shown in Figure 13, including the influences of the lifting
devices on the wake behaviors. However, the increased spacing allows for clearer observation of the spatial development
of the MRSLs' wakes. Notably, the in-plane velocity vectors indicate that the swirling motions persist even at more than
$9D$ downstream from MRSLs. This sustained vortical structure highlights the effectiveness of the lifting devices in wake
manipulation over long distances, thereby highlighting the efficacy of RGWF.

A closer examination of case **WL-10** in Figure 14 reveals a relatively sharp velocity variation at around $x/D = 5$. Note
that no MRSL are positioned at that location. This discontinuity can be attributed to the fact that flow data before and after
$x/D = 5$ are obtained from separate measurement sessions. Between these sessions, the scaled RGWFs are dismantled and
rebuilt, and the traverse system is manually repositioned (see Section 2.8). This observation suggests that the primary source
of experimental uncertainty is not the velocimetry itself, but rather the variability introduced during the reconstruction of
RGWFs. Nevertheless, it is important to point out that these uncertainties have minimal influence on the overall conclusions
of this study, as the aerodynamic effects of the lifting devices are sufficiently pronounced to remain clearly distinguishable
despite such imperfections.

Lastly, it is worth noting that the velocity contours obtained in this study closely resemble with the numerical results reported
by Li et al. (2025b) and Li et al. (2025c). This strong agreement not only validates the numerical simulations using experimental
data but also enhances the credibility of the present experimental findings, demonstrating that consistent conclusions about
RGWFs can be drawn from independent methodologies.


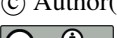


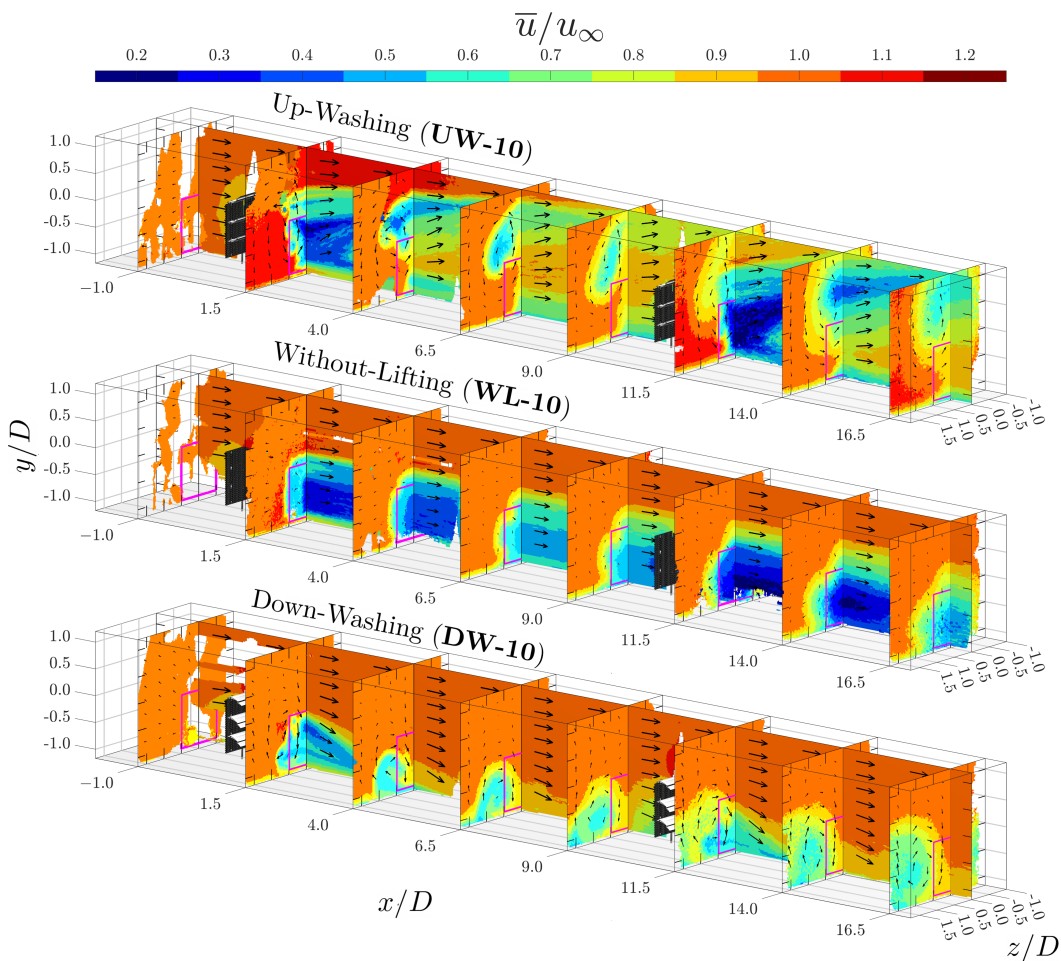

**Figure 14.** Holistic view of the time-averaged streamwise velocity ($\overline{u}$) fields for cases with a row spacing of $10D$, where MRSLs are positioned at $x/D = 0.0$ and $10.0$. The projected areas of the MRSLs are marked with magenta squares. Arrows indicate the direction of in-plane velocities, with their lengths scaled according to the in-plane velocity magnitudes.





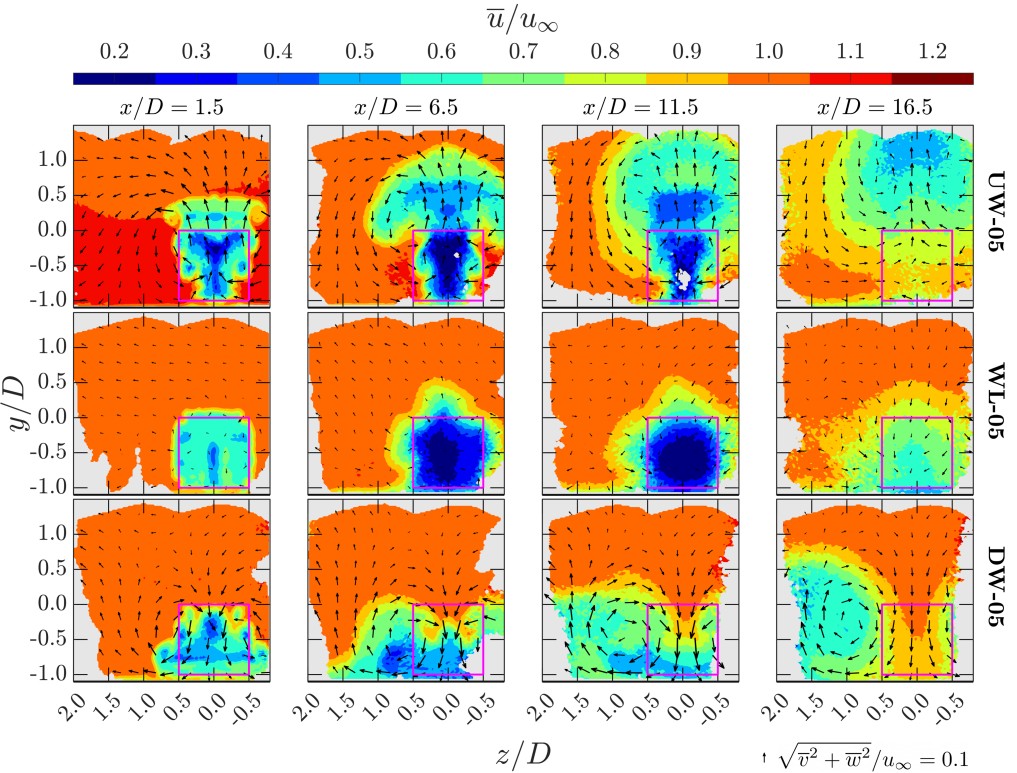

**Figure 15.** Planar views of the time-averaged streamwise velocity ($\overline{u}$) for cases with a row spacing of $5D$, shown at several $x$-planes. Case labels are indicated on the right side of each row, and the corresponding $x$-positions are labeled at the top of each column. The projected areas of the MRSLs are indicated by magenta squares. Arrows represent the direction of in-plane velocities, with their lengths scaled by their velocity norms. The scaling reference is shown at the bottom right of the figure.

### 3.3 Turbulence intensity

The contours of turbulence intensity (TI), defined in Equation (4), are shown in Figure 16 for cases with $\Delta_R = 5D$. TI fields are of particular interest, as turbulence is a dominant factor influencing the fatigue loads experienced by wind turbine systems
(Manwell et al., 2010; Watson et al., 2019). Moreover, previous studies have shown that turbulence significantly affects the large-scale coherent structures within wind turbine wakes, often diminishing their influence on downstream wake aerodynamics (Li et al., 2024; Yen et al., 2024). This effect is especially critical for RGWFs, which rely on the preservation of large-scale vortical structures to enhance vertical energy entrainment. Therefore, a thorough understanding of the role of turbulence is essential before pursuing practical implementations of the proposed concept.

$$\text{TI} \triangleq \frac{\sqrt{2k/3}}{u_\infty}, \qquad k \equiv \text{TKE} \triangleq \frac{1}{2}\left(\sigma_u^2 + \sigma_v^2 + \sigma_w^2\right) \tag{4}$$



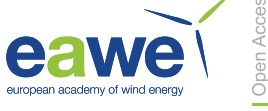

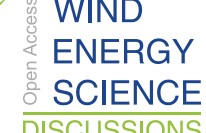

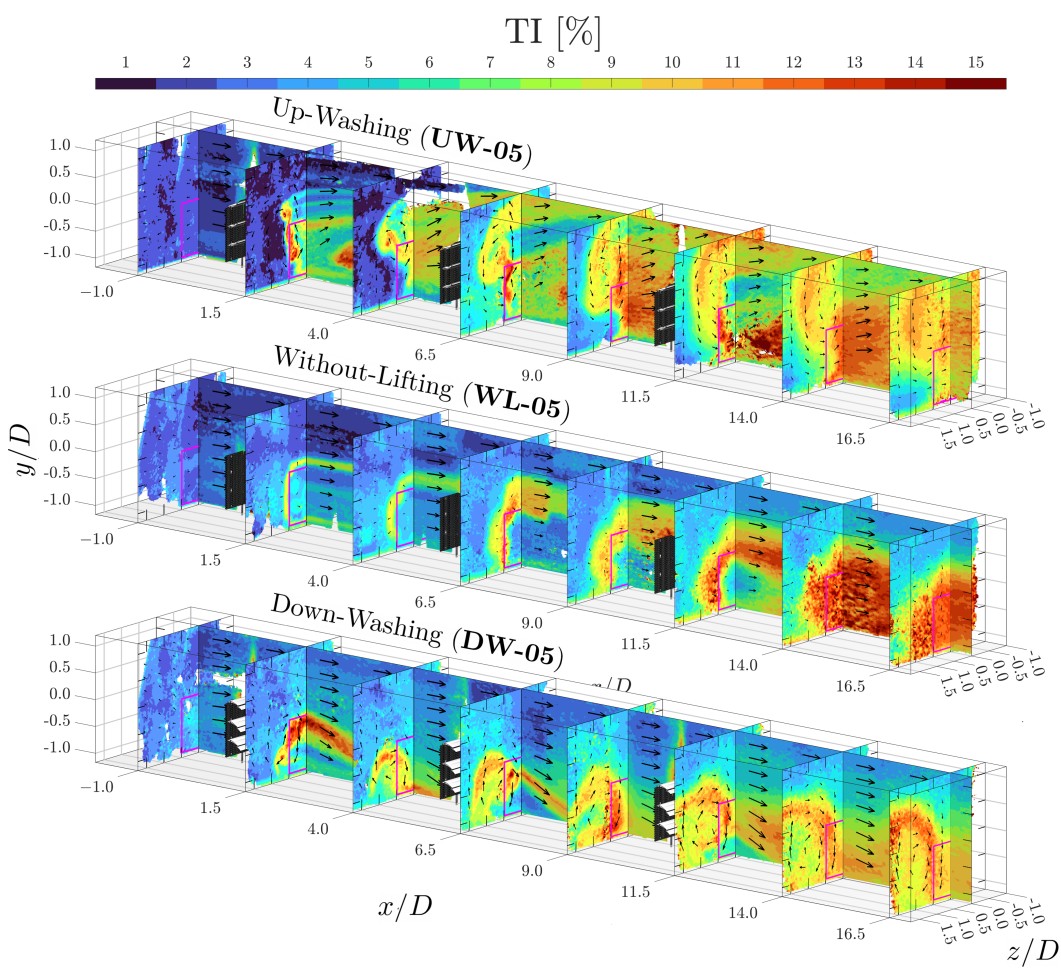

**Figure 16.** Holistic view of the turbulence intensity (TI) fields for cases with a row spacing of $5D$, where MRSLs are positioned at $x/D = 0.0$, $5.0$, and $10.0$. The projected areas of the MRSLs are marked with magenta squares. Arrows indicate the direction of in-plane velocities, with their lengths scaled according to the in-plane velocity magnitudes.





In general, the turbulence levels are observed to progressively increase as the flow passes through successive rows of MRSLs, consistent with findings from conventional wind turbine arrays (Wu and Porté-Agel, 2015). Specifically, prior to entering the domain of RGWF, the freestream turbulence intensity are found to be around 2% based on the values reported at $x/D = -1.0$. After passing the 1$^{st}$ and 2$^{nd}$ rows of MRSLs, TI exceeds 10% at $x/D = 4.0$ and reaches over 12% at $x/D = 9.0$, surpassing

typical offshore TI levels, which range from approximately 5% to 8% (Hansen et al., 2012).

Notably, vertical flows induced by the lifting devices remain significant beyond $x/D = 9.0$ for cases with $\Delta_R = 5D$ despite the presence of high levels of turbulence in this region, which can be seen in Figures 13. This indicates that the effectiveness of the MRSLs persists even under highly turbulent conditions. Furthermore, based on the thrust results in Figure 11, the performance of the 3$^{rd}$-row MRSLs equipped with lifting devices remains substantially higher than that of the configuration

without lifting devices. This implies that the vertical energy entrainment between the 2$^{nd}$ and 3$^{rd}$ rows is still significantly enhanced by the lifting devices despite the 2$^{nd}$-row MRSLs are subjected to highly-turbulent flows. These experimental findings support the numerical results of Li et al. (2025b), who reported that inflow turbulence has minimal impact on the aerodynamic performance of MRSL lifting devices.

While the current results preliminarily show that the MRSL concept is relatively insensitive to ambient turbulence, further

experiments with controlled inflow turbulence levels are necessary to obtain more quantitative insights. Such data would be critical for translating the proposed concept into real-world applications.

Furthermore, Figure 16 reveals several features of the MRSL wake structures. Specifically, three sets of regions exhibiting elevated turbulence intensity are identified. The first set corresponds to the wake edges, which is attributed to the strong velocity shear (Wu and Porté-Agel, 2011). The second set are the regions of the swirling centers, which are clearly visible

when examining the TI contours in Figure 16 alongside the vorticity contours in Figure 17. These strong velocity fluctuations are due to the instabilities of the tip vortices (Giuni and Green, 2013). The third set of regions corresponds to the wakes of the wings, most clearly observed in case **UW-05** as three distinct strips with high values of TI can be identified on the $z/D = 0.0$ plane near $x/D = 2.5$. These high TI stripes are resulted from the shear layer instabilities developed along and after the wings (Wang and Ghaemi, 2022). In general, the locations and characteristics of these high-TI regions are consistent with prior

findings on the general features of wake flow, which reinforce the validity of the present experimental results.

### 3.4 Time-averaged streamwise vorticity

This section examines the vorticity fields to investigate the tip vortices generated by the lifting devices of MRSLs. These vortical structures are closely associated with the swirling motions previously observed in Figures 13 and 14. Contours of the time-averaged streamwise vorticity $\overline{\omega}_x$ at several $x$-planes for cases with $\Delta_R = 5D$ are presented in Figure 17. A more

comprehensive set of contour plots is available in Fijen (2025), including those for the cases with $\Delta_R = 10D$.

To reduce noise in the vorticity fields, the data are spatially filtered by averaging over a spherical volume with a radius of $0.06D$, which is three times the binning size. The averaging is weighted by the particle count $N$. This filtering procedure is applied only in this section and in Section 3.6, where contours of spatial derivatives are studied.



WIND
ENERGY
SCIENCE
DISCUSSIONS

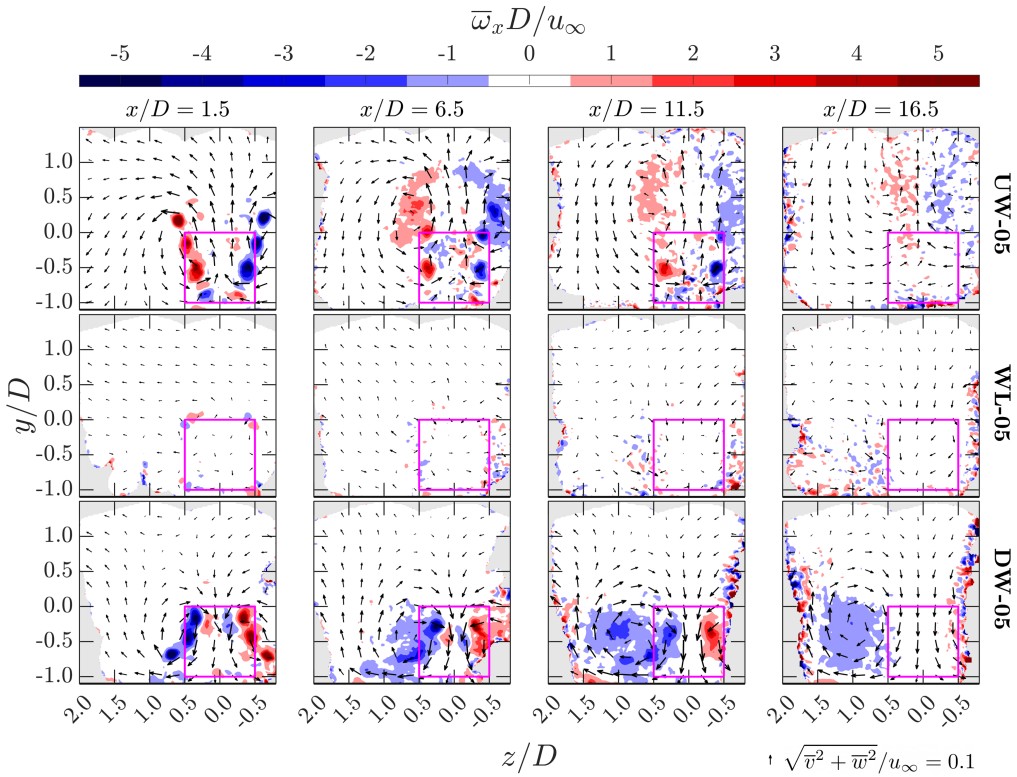

**Figure 17.** Contours of time-averaged streamwise vorticity ($\overline{\omega}_x$) for cases with a row spacing of $5D$ at several $x$-planes. Case labels are indicated on the right side of each row, and the corresponding $x$-positions are labeled at the top of each column. The projected areas of the MRSLs are marked with magenta squares. In-plane velocity directions are illustrated by arrows, scaled according to their magnitudes. The reference scale for the arrows is provided in the bottom-right corner of the figure. Note that the data have been filtered by spatial averaging over a spherical volume with a radius of $0.06D$.

In Figure 17, no significant vortical structures are observed throughout the RGWF in the case without lifting devices (**WL-05**). This outcome is expected, as no significant circulations are induced.

In the up-washing case (**UW-05**), three pairs of tip vortices generated by the wings of the 1[st]-row MRSL are clearly identifiable at $x/D = 1.5$. By $x/D = 6.5$, these tip vortices have diffused, forming broader regions of vorticity with reduced magnitude. At the same streamwise location, three new pairs of tip vortices released by the wings of the 2[nd]-row MRSL can be observed, overlapping onto the remnants vortices of the 1[st]-row MRSL. This observation supports the findings of Li et al. (2025c), who reported that the circulation system within RGWFs can be accumulated through successive rows of MRSLs. At $x/D = 11.5$, the individual tip vortices from the 3[rd]-row MRSL become less distinguishable, with only the vortices from the bottom wing remain clearly outlined. Besides the accumulated turbulence within the wind farm (see Figure 16), this degradation may also result from the drop in $L^W$ due to the strong vertical flow induced by the upstream MRSLs, which makes the downstream MRSLs to operated at off-design conditions as changes the in angle of attack are significant (see the in-plane ve-





locity vectors in Figure 17). As stated in Section 3.1, this finding again suggests that the effectiveness of MRSL lifting devices could be further improved by enabling active control of wing pitch angles or by applying the wings that are less sensitive to variations in angel of attack. By $x/D = 16.5$, the streamwise vortical structures have largely dissipated, which is ascribed to the large turbulent fluctuations downstream of the 3$^\text{rd}$ row, as shown in Figure 16.

In the down-washing case (**DW-05**), the tip vortices from the three wings of the 1$^\text{st}$-row MRSL are clearly visible at $x/D = 1.5$, similar to those observed in the up-washing case. As with **UW-05**, the vortices from the 2$^\text{nd}$-row MRSLs overlap with the diffused vortical structures originating from the 1$^\text{st}$ row. Interestingly, unlike the up-washing configuration, the tip vortices from all three wings of the 3$^\text{rd}$-row MRSL in case **DW-05** remain identifiable at $x/D = 11.5$. Moreover, at $x/D = 16.5$, the vortical structures in the down-washing case appear stronger than those in the up-washing case. The underlying cause of this difference is not explored in the present study. One plausible explanation is the higher lift forces ($|L^W|$) generated by the MRSLs in the 2$^\text{nd}$ and 3$^\text{rd}$ rows of **DW-05** compared to those in **UW-05**. Another possibility is the presence of the floor, which may interact more directly with the downward-directed wake and thereby affect the vortical structures.

Another aspect worth highlighting is the vertical positions of the swirling centers. In the up-washing case, the swirling centers extend from around the center of the MRSL projection area ($y \simeq -0.5$) upward into the higher regions ($y/D > 0.5$) as the flow progress deeper into RGFW. This behavior is closely linked to the self-propelling nature of vortex pairs released by MRSLs, which is elaborated in the work of Avila Correia Martins et al. (2025). In contrast, for the down-washing case, the swirling centers remain confined below the top of MRSLs. This distinction may make the up-washing configuration more favorable than the down-washing configuration, given that the primary objective of MRSL and RGWF is to enhance vertical energy entrainment. As illustrated with the velocity vectors in Figures 13 and 17, compared to the down-washing configuration, the up-washing configuration demonstrates a greater capacity to mix the MRSL wake with the higher-altitude flow.

### 3.5 Available power based on the frontal projection area

Figure 18 presents the available power averaged over the MRSL's frontal projection area, denoted as $<\overline{u}^3>_\text{PA}$, along the streamwise direction. The definition of $<\overline{u}^3>_\text{PA}$ used in this study is provided in Equation (5). During the averaging process, unavailable data points (NaNs) are excluded. $<\overline{u}^3>_\text{PA}$ is of particular interest because it serves as an indicator of the integral characteristics of the MRSL wake, directly indicating the flow energy available for harvesting. As such, it is used in this study as a representative metric for evaluating the overall wake recovery rate.

$$<\overline{u}^3>_\text{PA}(x) \triangleq \frac{\int_{-0.5D}^{0.5D} \int_{-1.0D}^{0.0D} [\overline{u}(x)]^3 \, \mathrm{d}y \, \mathrm{d}z}{D^2} \tag{5}$$

In Figure 18, differences in $<\overline{u}^3>_\text{PA}$ are already apparent upstream of the 1$^\text{st}$-row MRSLs. For $x/D < 0.0$, the cases with up-washing configuration exhibit higher values of $<\overline{u}^3>_\text{PA}$ compared to those with without-lifting configuration, while those with down-washing configuration show the lowest values among the three configurations. These trends are consistent with the thrust forces ($T^R$) reported in Figures 11 and 12. Also, the aerodynamic mechanisms that account for the variations in $T^R$ for the 1$^\text{st}$-row MRSLs again explain the observed differences in $<\overline{u}^3>_\text{PA}$ at $x/D < 0.0$.





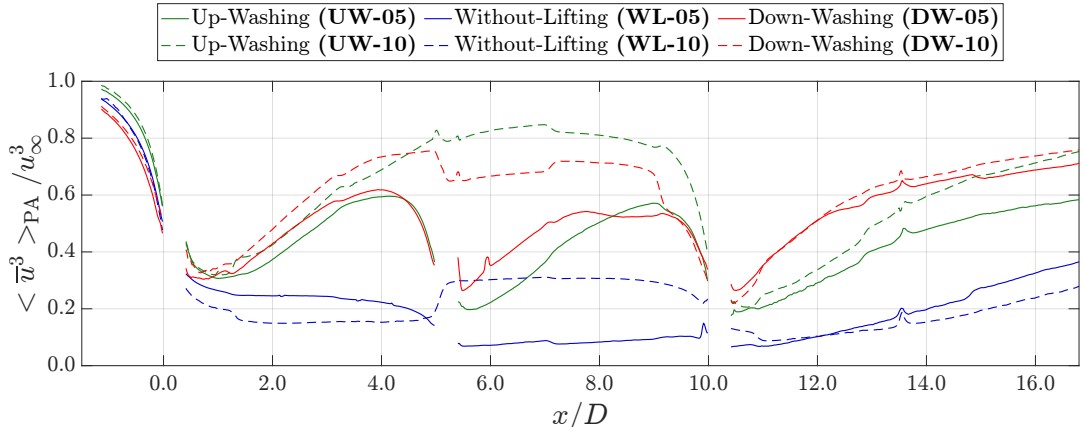

**Figure 18.** Available power, $< \overline{u}^3 >_{\text{PA}}$, for RGWFs with their MRSLs configured differently. Results are shown for both $\Delta_R = 5D$ (solid lines) and $\Delta_R = 10D$ (dashed lines), with case labels corresponding to those introduced in Table 1. MRSLs are located at $x/D = 0.0$, $5.0$, and $10.0$ for $\Delta_R = 5D$, and at $x/D = 0.0$ and $5.0$ for $\Delta_R = 10D$. The available power is computed using Equation (5).

In general, all six cases follow the one-dimensional momentum theory (Manwell et al., 2010), where the $< \overline{u}^3 >_{\text{PA}}$ curves decrease with increasing $x$ immediately upstream and downstream of the 1st-row MRSLs at $x/D = 0.0$. Subsequently, around $x/D = 1.0$, the available power in the up-washing and down-washing cases begins to increase, indicating the onset of wake

recovery. In contrast, for without-lifting configuration, $< \overline{u}^3 >_{\text{PA}}$ remains nearly constant, suggesting that wake recovery is absent or minimal. The trends in $< \overline{u}^3 >_{\text{PA}}$ closely correspond to the velocity fields shown in Figures 13 and 14, which demonstrate significant wake recovery for up-washing and down-washing cases, but not for without-lifting case.

At $x/D = 5.0$ for cases with $\Delta_R = 10D$, the available power for cases **UW-10** and **DW-10** are approximately three to four times greater than that for case **WL-10**. This trend aligns well with the thrust measurements for the cases with $\Delta_R = 5D$ shown

in Figure 11, where the 2nd-row MRSLs equipped with lifting devices exhibit $T^R$ that is around three times of that without.

The profiles of $< \overline{u}^3 >_{\text{PA}}$ around the 2nd-row MRSLs for the cases with $\Delta_D = 5D$ generally follow a similar trend to those at the 1st-row. A drop in $< \overline{u}^3 >_{\text{PA}}$ is again observed immediately upstream and downstream of the 2nd-row MRSLs at $x/D = 5.0$. After the flow passes the 2nd-row MRSLs, the cases with up-washing and down-washing configurations exhibit strong wake recovery, in sharp contrast to the minimal recovery observed in those case with without-lifting.

The profiles of $< \overline{u}^3 >_{\text{PA}}$ after the 3rd-row MRSLs for the cases with $\Delta_D = 5D$ and those after the 2nd-row MRSLs for those with $\Delta_D = 10D$ display similar trends. In general, all profiles show evidence of wake recovery. Notably, even the cases with without-lifting configuration exhibit recovery in these regions, primarily due to the strong turbulent mixing after $x/D = 10$, as shown by fields of turbulence intensity presented in Figure 16. A closer examination of the slopes of $< \overline{u}^3 >_{\text{PA}}$ reveals that up-washing and down-washing configurations again achieve significantly stronger wake recovery than without-lifting. This

result demonstrates that the lifting devices remain effective even when MRSLs operate within the highly turbulent wakes of the





upstream ones. This finding once more supports the conclusion that the design of MRSL is resilient to turbulence, in agreement with the statement in Section 3.3 and the numerical results reported by Li et al. (2025b).

A closer examination of Figure 18 reveals several unexpected jumps in $< \overline{u}^3 >_{\mathrm{PA}}$ along the streamwise direction, highlighting certain accuracy limitations of the present experiments. These spurious jumps are particularly evident around $x/D = 5$ in

the $\Delta_D = 10D$ cases, coinciding with the boundary between two traverse positions, which likely contributes to the observed irregularities. As detailed in Section 2.8, the regenerative wind farms are repeatedly disassembled and reassembled during the experimental campaign, which can introduce some inconsistencies. Additionally, minor spikes in $< \overline{u}^3 >_{\mathrm{PA}}$ stem from movable wooden elements (see Figure 2) that occasionally block the light source, contaminating the data. Despite these measurement imperfections, current results clearly demonstrate that both up-washing and down-washing configurations substantially outper-

form the configuration without lifting devices in terms of overall wind farm performance and wake recovery rate, which are the central conclusions of this study.

### 3.6 Analysis of wake recovery based on the re-distribution of mean kinetic energy

In this section, the wake recovery within RGWF is analyzed based on the re-distribution of the mean kinetic energy. Before starting the analysis, an explanation of the subscript $\hat{z}$ is provided and the filtering of the data is described. The subscripts

$\hat{z}$ in this work indicate that the properties are averaged over two points that are opposite with respect to the symmetry plane $z/D = 0$, as detailed in Equation (6). This in fact corresponds to mirroring the half of the flow field across the symmetry plane $z/D = 0$ and average it with the another half of the flow fields. The $\pm$ sign in the equation becomes minus only when property $B$ is the $z$-component of a vector. Note that when averaging the experimental data, the data are weighted by the particle counts $N$ ($N = 1$ is set everywhere for LES data). Additionally, in order to reduce the noise, the experimental data in this part are

filtered by spatially averaging over a spherical volume with a radius of $0.06D$, where the procedure is same as that for $\overline{\omega}_x$ in Section 3.4.

$$\text{For property } B(x,y,z): \quad B_{\hat{z}}(x,y,\hat{z}) \equiv \frac{[N(x,y,\hat{z}) \times B(x,y,\hat{z}) \pm N(x,y,-\hat{z}) \times B(x,y,-\hat{z})]}{[N(x,y,\hat{z}) + N(x,y,-\hat{z})]} \tag{6}$$

#### 3.6.1 Energy re-distribution for wake recovery

To investigate the recovery process of the MRSLs' wakes in greater details, this part focuses on the term $\partial \overline{u} K / \partial x$ (abbreviated

as $\mathbb{A}$), which represents the spatial changing rate (SCR) of the mean kinetic energy (MKE) flux in the streamwise direction. Here, mean kinetic energy is defined as $\mathrm{MKE} \equiv K \overset{\Delta}{=} 0.5 \overline{u}_i \overline{u}_i$. A positive value of $\mathbb{A}$ indicates an increase in streamwise energy flux along the $x$-direction. Figure 19 presents contour plots of $\mathbb{A}$ on the left side of each panel for cases with $\Delta_R = 5D$ at several $x$-planes.

In Figure 19, it is evident that the cases equipped with the lifting devices exhibit significantly positive values of $\mathbb{A}$ within

the MRSLs' projection areas, indicating substantial wake recovery. In contrast, large regions with negative $\mathbb{A}$ values appear outside the projection areas in cases **UW-05** and **DW-05**, highlighting areas from which energy is redistributed to accelerate





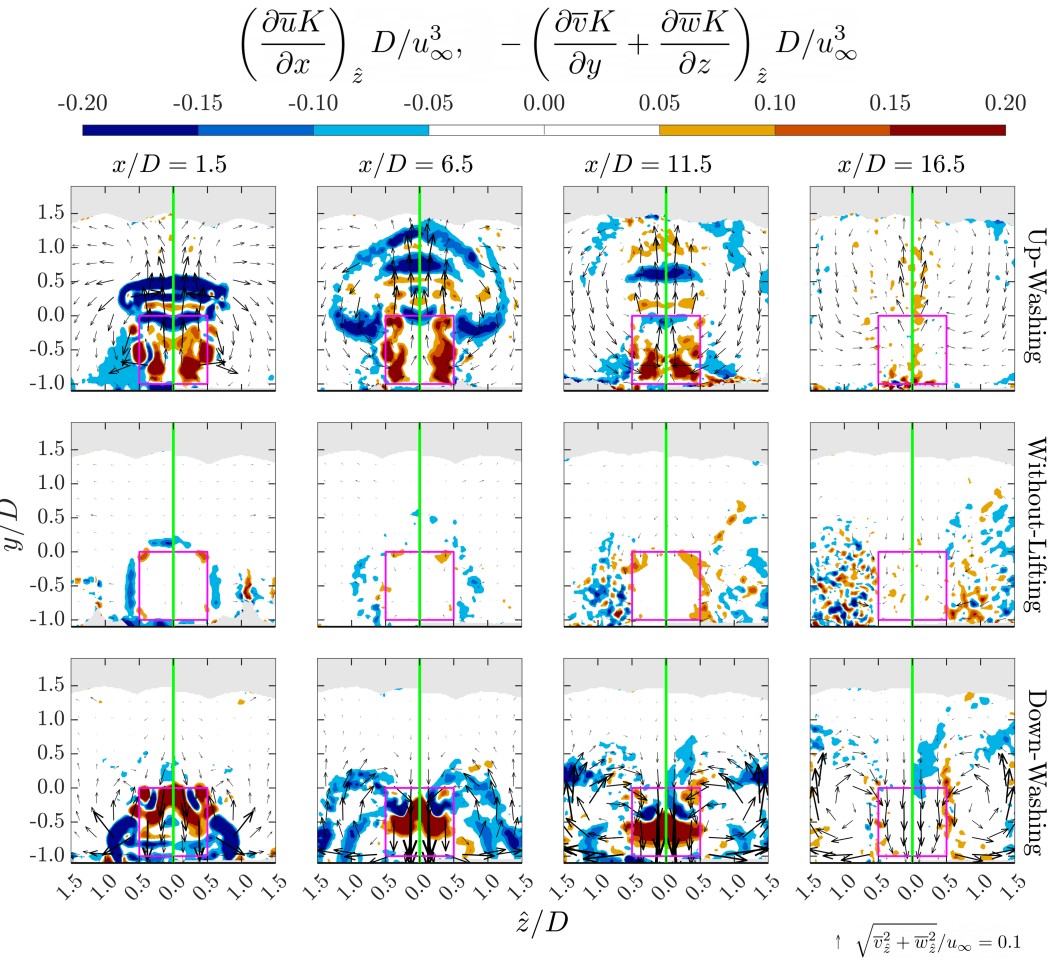

**Figure 19.** Contours of the terms related to energy redistribution, as defined in Equation (7), for cases with a row spacing of $5D$. The term $\partial \overline{u}K/\partial x$ is shown on the left side of each panel, while $-(\partial \overline{v}K/\partial y + \partial \overline{w}K/\partial z)$ is shown on the right. Cases **UW-05**, **WL-05**, and **DW-05** are displayed in the top, middle, and bottom rows, respectively. The $x$-positions of the corresponding planes are indicated at the top of each column. Subscript $\hat{z}$ denote that the values of the property are averaged over the symmetry plane $z/D = 0$ (see Equation (6)). In-plane velocity directions are represented by arrows scaled by their magnitude, with the reference scale provided in the bottom-right corner. Note that all spatial derivatives data are filtered by averaging over a spherical volume with the radius being $0.06D$.





wake recovery. Moreover, some regions with negative $\mathbb{A}$ for cases **UW-05** and **DW-05** are above the top of the MRSL ($y > 0.0$). This confirms that the lifting devices enhance vertical entrainment by channeling flow energy from higher altitudes downward. This effect is especially pronounced in the case with up-washing configuration, where the negative $\mathbb{A}$ patches extend beyond

the measurement domain after the 3rd-row MRSL. This observation aligns well with the velocity contours shown in Figure 13, which reveal that the wake in case **UW-05** is effectively ejected into higher layers.

In contrast, compared to the cases equipped with lifting devices, regions with notable $\mathbb{A}$ magnitudes are considerably smaller for case **WL-05**, indicating weak vertical mixing and limited energy redistribution. This pattern clearly reflects the slow wake recovery rate that conventional wind farms currently suffered.

To further investigate the mechanism of energy redistribution reflected by the term $\partial \overline{u} K / \partial x$, Equation (7) is utilized. This equation is derived by rearranging the transport equations of time-averaged mechanical power for incompressible flow. The terms $\partial \overline{v} K / \partial y$ and $\partial \overline{w} K / \partial z$ represent the spatial changing rates (SCR) of mean kinetic energy due to vertical and lateral advection, respectively. Contributions from second and third order Reynolds stresses, pressure gradients, and viscous effects are collectively grouped into the residual term $\mathbb{R}$.

$$\underbrace{\frac{\partial \overline{u} K}{\partial x}}_{\text{SCR of MKE flux in } x\text{-direction } (\mathbb{A})} = - \left( \underbrace{\frac{\partial \overline{v} K}{\partial y}}_{\text{SCR of MKE flux in } y\text{-direction } (\mathbb{B})} + \underbrace{\frac{\partial \overline{w} K}{\partial z}}_{\text{SCR of MKE flux in } z\text{-direction } (\mathbb{C})} \right) + \underbrace{\mathbb{R}}_{\text{residuals}} \qquad (7)$$

The contributions of the SCR of MKE in the $y$- and $z$-directions, denoted by $-(\mathbb{B}+\mathbb{C})$, are shown on the right side of each panel in Figure 19. Comparison with the corresponding $\mathbb{A}$ distributions on the left sides of the panels reveals that the residual term $\mathbb{R}$ is generally small, especially for the cases with lifting devices. The low magnitudes of $\mathbb{R}$ suggest that energy redistribution in the RGWFs studied is primarily governed by advection processes, such that $\mathbb{A} \simeq -(\mathbb{B}+\mathbb{C})$. This finding is

consistent with previous numerical studies (Li et al., 2025b, c). Importantly, this indicates that the wake recovery mechanism in RGWFs fundamentally differs from that of conventional wind farms. While RGWFs rely mainly on advective transport to redistribute energy, conventional wind farms depend more heavily on the processes driven by Reynolds stresses (Calaf et al., 2010; Porté-Agel et al., 2020). (Recall that the contributions of Reynolds stresses are accounted by the residual term $\mathbb{R}$.)

### 3.6.2 Comparing with the numerical results

This part compares the experimental results of the present study with the numerical outcomes of Li et al. (2025b), who conducted large eddy simulations (LES) using actuator methods to investigate the wake aerodynamics of stand-alone MRSL with different configurations of lifting devices. Specifically, cases **UW-10**, **WL-10**, and **DW-10** from the current experiments are compared with cases **UW-17**, **WL-17**, and **DW-17** from their simulations. The suffixes refer to the row spacing and the inflow turbulence intensity used in each study, respectively. For the purpose of this comparison, the flow fields of the experimental

cases with $\Delta_R = 10D$ can be considered representative of stand-alone MRSL behavior for $x/D < 9.0$, as the MRSLs in the 2nd row have negligible to limited influence.



The numerical setup in Li et al. (2025b) is broadly comparable to the present experiments. Particularly, the ground clearance of MRSL is $0.10D$, the thrust coefficient $C_T$ for the configuration without lifting devices is $0.71$, the normalized lift force $\widehat{L}^W$ (see Equation (2)) is approximately $100\%$, the inflow is without vertical velocity shear, and the inflow turbulence intensity is $1.74\%$. However, key differences exist. Their simulations use full-scale MRSLs with a side length of $D = 300$ m and a freestream velocity of $u_\infty = 10.1$ m/s, resulting in a Reynolds number $Re_D \stackrel{\Delta}{=} u_\infty D/\nu$ of approximately $2.0 \times 10^8$. In contrast, the Reynolds number in the current experiments is around $1.4 \times 10^5$, leading to a significant mismatch in flow regimes. Additionally, the simulated MRSLs in Li et al. (2025b) are equipped with four wings rather than three, and the floor boundary conditions are modeled as a slip wall, unlike the physical floor present in the experiments. For further details on their numerical setup, readers are referred to Li et al. (2025b).

Figure 20 presents contour plots of $\partial \overline{u} K / \partial x$ from both the current experiments and the LES results reported by Li et al. (2025b). In each panel, the experimental data are shown on the left, and the corresponding LES results are shown on the right. Overall, the experimental and numerical results show strong agreement across all three configurations, despite the significant discrepancies in Reynolds numbers $Re_D$ and some differences in experimental setup and numerical modeling. Both datasets consistently demonstrate that the up-washing and down-washing configurations exhibit substantially stronger wake recovery compared to the configuration without lifting devices. In particular, for the up-washing configuration, both results indicate that wake recovery is primarily driven by the redistribution of flow energy from upper to lower layers. These consistencies reinforce the conclusion that the MRSL concept significantly enhance vertical energy entrainment and hold strong potential for improving overall wind farm efficiency.

Although the current experimental results align well with the numerical findings of Li et al. (2025b), several subtle differences are observed. Notably, in the experiments, wake development appears to occur more rapidly in the cases with lifting devices compared to the simulations. This discrepancy is likely due to the distribution of lift forces ($L^W$) being more concentrated on the top wing of the MRSL in the experiments. In the experimental setup, the top wing is exposed to undisturbed flow, while the lower wings operate within the wake of the porous disk, resulting in reduced lift (see Appendix B2). In contrast, the numerical simulations model all wings as overlapping with the actuator disk, leading to a more uniform distribution of lift among them. This difference likely causes the MRSLs in the experiments to generate circulation systems concentrated at higher vertical positions. As a result, the induced swirling motions become stronger, and this mechanism may explain why the experimental wakes develop faster than those observed in the simulations.

## 3.7 RGWF with the staggered layout

This section addresses the aerodynamics of RGWFs when MRSLs in different rows are not aligned with the streamwise direction. Understanding this aspect is essential for bringing RGWF from concept to practical applications. In real-world scenarios, while prevailing wind directions may exist, wind directions at wind farm sites vary over time. As a result, if MRSLs always face against the inflow direction, the effective layout of an RGWF continuously changes. Previous studies have shown that wind farm layout significantly influences the power output of conventional wind farms (Barthelmie et al., 2010; Stevens et al., 2016). For RGWFs, layout sensitivity is postulated to be even more pronounced. This expectation is supported by the


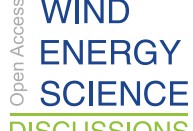


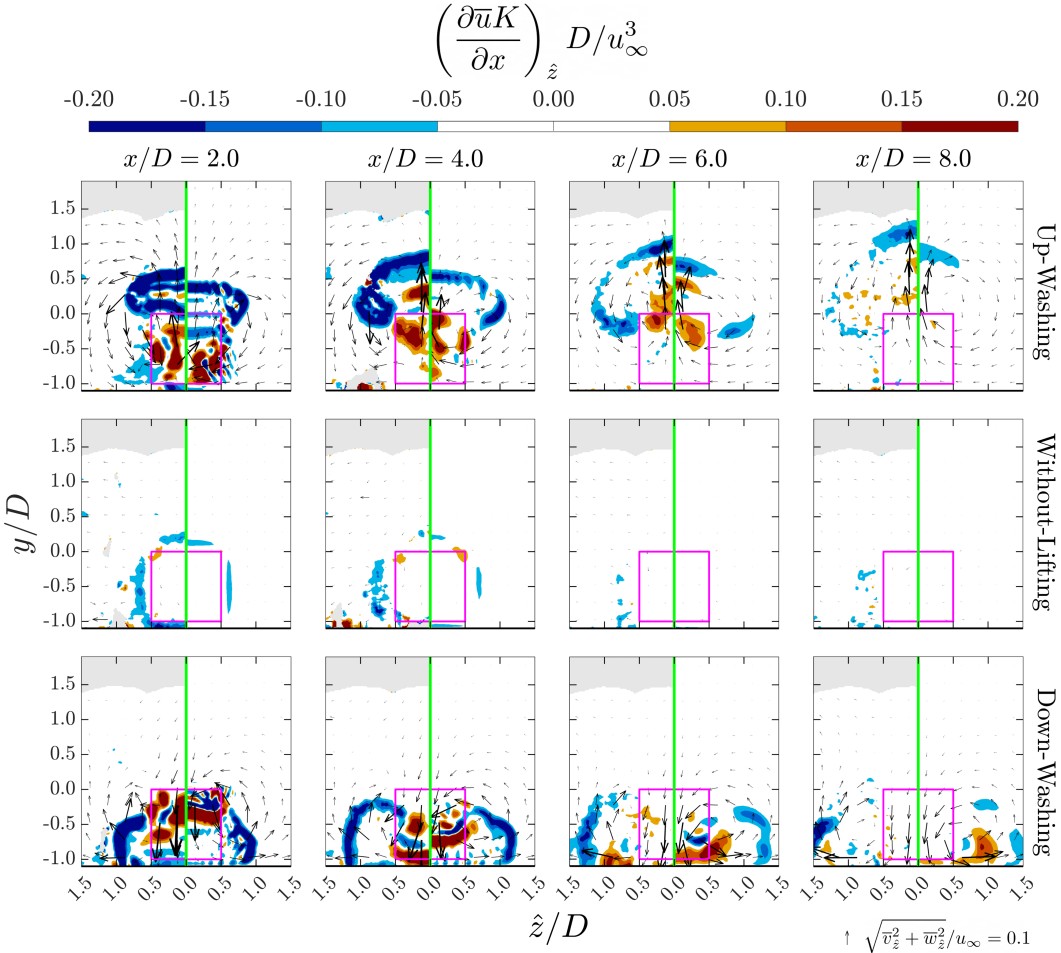

**Figure 20.** Contours of $\partial \overline{u}K/\partial x$, as defined in Equation (7), for cases with a row spacing of $10D$ (**UW-10**, **WL-10**, and **DW-10**), along with the corresponding numerical results from Li et al. (2025b). See the main text for the key parameters of their numerical setup. In each panel, experimental results are shown on the left, and LES results from Li et al. (2025b) are shown on the right. The up-washing, without-lifting, and down-washing configurations are displayed in the top, middle, and bottom rows, respectively. The $x$-positions of the corresponding planes are indicated at the top of each column. Subscript $\hat{z}$ denote that the values of the property are averaged over the symmetry plane $z/D = 0$ (see Equation (6)). In-plane velocity directions are represented by arrows scaled by their magnitude, with the reference scale provided in the bottom-right corner. Note that all spatial derivatives data from the experiments are filtered by averaging over a spherical volume with the radius being $0.06D$.





discussion in Section 3.4, which highlights that enhanced vertical energy entrainment in RGWFs originates from streamwise vortical structures that accumulate strength progressively when MRSLs are arranged in a streamwise-aligned layout.

To assess whether the effectiveness of RGWF is sensitive to wind farm layout, a case with a staggered wind farm layout is tested, which is case **UW-05-ST** listed in Table 1. The case features two rows of MRSLs that are mis-aligned with the inflow

direction. The detailed dimensions of the RGWF layout for this case are given in Section 2.3.

Figures 21 to 23 present the contours of time-averaged streamwise velocity $\overline{u}$, time-averaged streamwise vorticity $\overline{\omega}_x$, and time-averaged vertical velocity $\overline{v}$ at several $x$-planes for the case with staggered wind farm layout. The in-plane velocity components, $\overline{v}$ and $\overline{w}$, are superimposed as arrows to illustrate the flow direction. For comparison, results from the aligned case **UW-05** are also shown alongside. However, it should be noted that at $x/D = 10.4$, direct comparison becomes less meaningful,

as case **UW-05** includes 3rd-row MRSLs, whereas case **UW-05-ST** does not.

As expected, at $x/D = 4.4$, which is the plane located just upstream of the 2nd-row MRSLs, the contours of $\overline{u}$, $\overline{\omega}_x$, and $\overline{v}$ are generally similar between cases **UW-05** and **UW-05-ST**. Minor differences may be attributed to the variations introduced while manually reassembling RGWFs.

At $x/D = 6.4$ and $8.4$, which are both downstream of the 2nd-row MRSLs, the effects of mis-alignment become pronounced.

In case **UW-05-ST**, Figure 21 shows that the wake of the 2nd-row-center-column MRSL is centered around $z/D = -1.0$, and the wake of the side-column MRSL located on the positive $z$ side is also visible. Examining the swirling motions depicted by the in-plane velocity vectors, the counter-rotating vortex pair in the staggered case appears asymmetric. Specifically, the vortex on the negative $z$ side is smaller in size, while the one on the positive $z$ side is larger. Although the vortex shapes are altered by the mis-alignment, the updraft and downdraft motions found in case **UW-05-ST** remain significantly stronger than those in

the without-lifting case, as evident from the in-plane velocity arrows in Figure 17. This suggests that the vertical entrainment remains robust in the staggered-up-washing case and is still substantially stronger than that in the without-lifting case.

Examining the contours of $\overline{\omega}_x$ for case **UW-05-ST** at $x/D = 6.4$ and $8.4$ in Figure 22, it is evident that vortical structures generated by the 1st-row-center-column MRSL and the 2nd-row side-column MRSL jointly contribute to downdraft motions at around $z/D = 1.0$. This observation indicates that mis-alignment between MRSLs increases the interaction between vortical

structures released by adjacent columns. Moreover, the vorticity contours show that positive $\overline{\omega}_x$ generated by the 2nd-row-center-column MRSL overlaps with negative $\overline{\omega}_x$ originating from the 1st-row-center-column MRSL. These interactions reduce the strength of the positive $\overline{\omega}_x$, ultimately causing the in-plane velocity field to skew toward the negative $z$-direction. Despite this attenuation of vorticity, the vertical flow induced by the lifting devices remains strong in case **UW-05-ST**, indicating that vertical entrainment continues to be significantly enhanced.

Although the contours of $\overline{u}$ and $\overline{\omega}_x$ suggest that the RGWF concept is not significantly weakened by the mis-alignment, the time-averaged vertical velocity $\overline{v}$ contours in Figure 23 indicate that vertical flows are somewhat less intense in the staggered layout compared to the aligned one. This conclusion is drawn from the $\overline{v}$ fields at $x/D = 6.4$ and $8.4$, which show that regions of strong vertical velocity penetrate to higher altitudes in case **UW-05** than in case **UW-05-ST**. Specifically, regions with $\overline{v}/u_\infty > 0.1$ can only be found below $y/D = 0.5$ for case **UW-05-ST** while those regins can be found above $y/D = 1.2$ for

case **UW-05**. This indicates that, in the staggered case, a thinner upper-layer flow contributes to wake re-energization, whereas



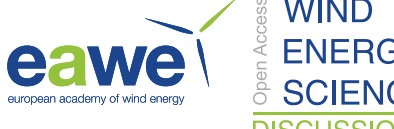

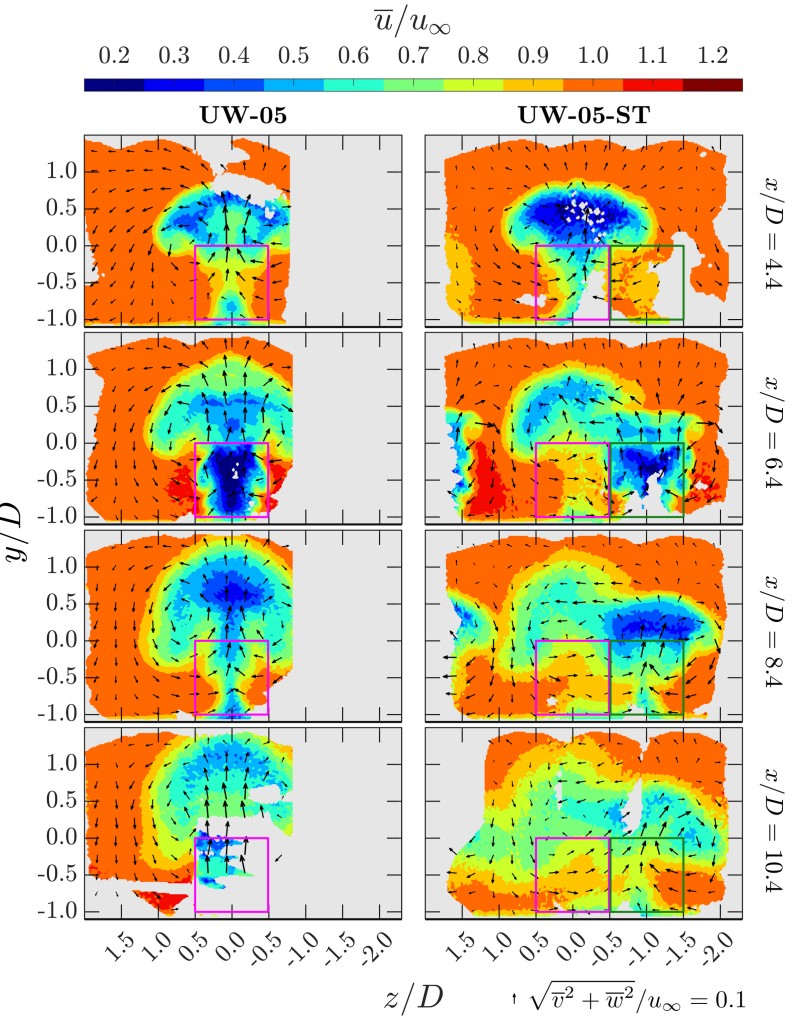

**Figure 21.** Contours of the time-averaged streamwise velocity $\overline{u}$ at selected $x$-planes for the two up-washing cases. The case with the aligned layout (case **UW-05**) and the staggered layout (case **UW-05-ST**) are presented in the left and right column, respectively. The corresponding $x$-positions are indicated on the right side of each row. For the aligned case, MRSL projection areas are marked with magenta squares. For the staggered case, the projection areas of the 1st- and 2nd-row MRSLs (see Figure 7) are shown in magenta and dark green squares, respectively. In-plane velocity directions are represented by arrows scaled by their magnitude, with the reference scale provided at the bottom-right corner of the figure.

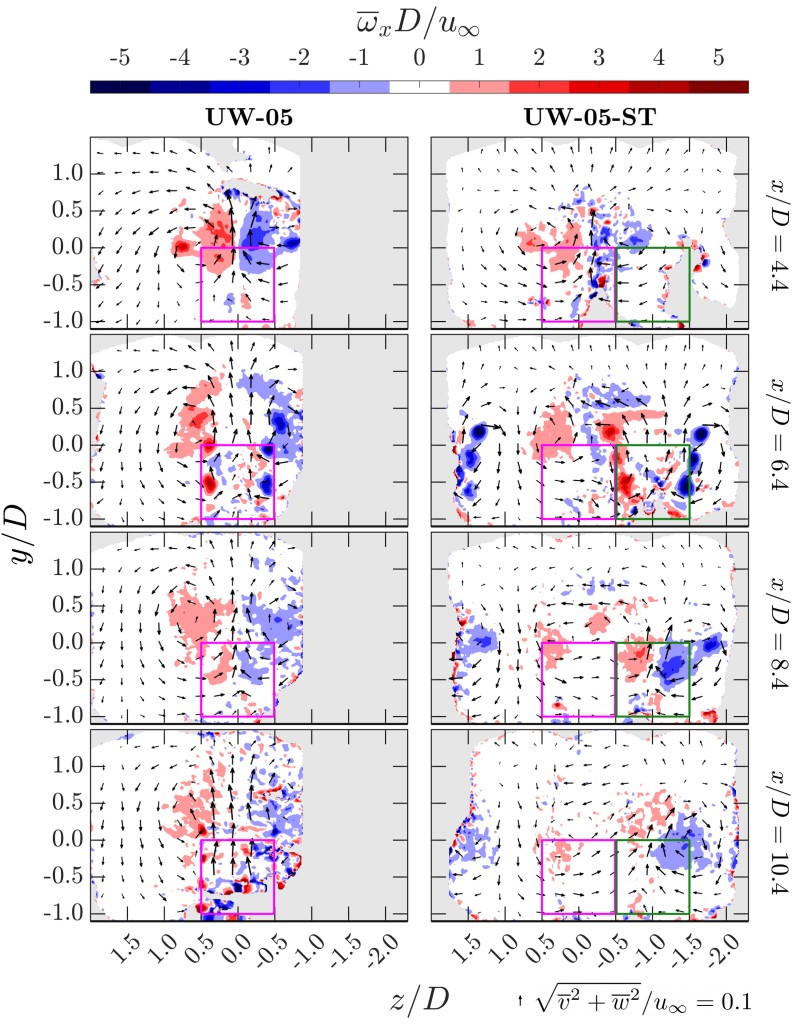

**Figure 22.** Contours of the time-averaged streamwise vorticity $\overline{\omega}_x$ at selected $x$-planes for the two up-washing cases. The case with the aligned layout (case **UW-05**) and the staggered layout (case **UW-05-ST**) are presented in the left and right column, respectively. The corresponding $x$-positions are indicated on the right side of each row. For the aligned case, MRSL projection areas are marked with magenta squares. For the staggered case, the projection areas of the 1[st]- and 2[nd]-row MRSLs (see Figure 7) are shown in magenta and dark green squares, respectively. In-plane velocity directions are represented by arrows scaled by their magnitude, with the reference scale provided at the bottom-right corner of the figure.

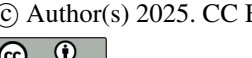

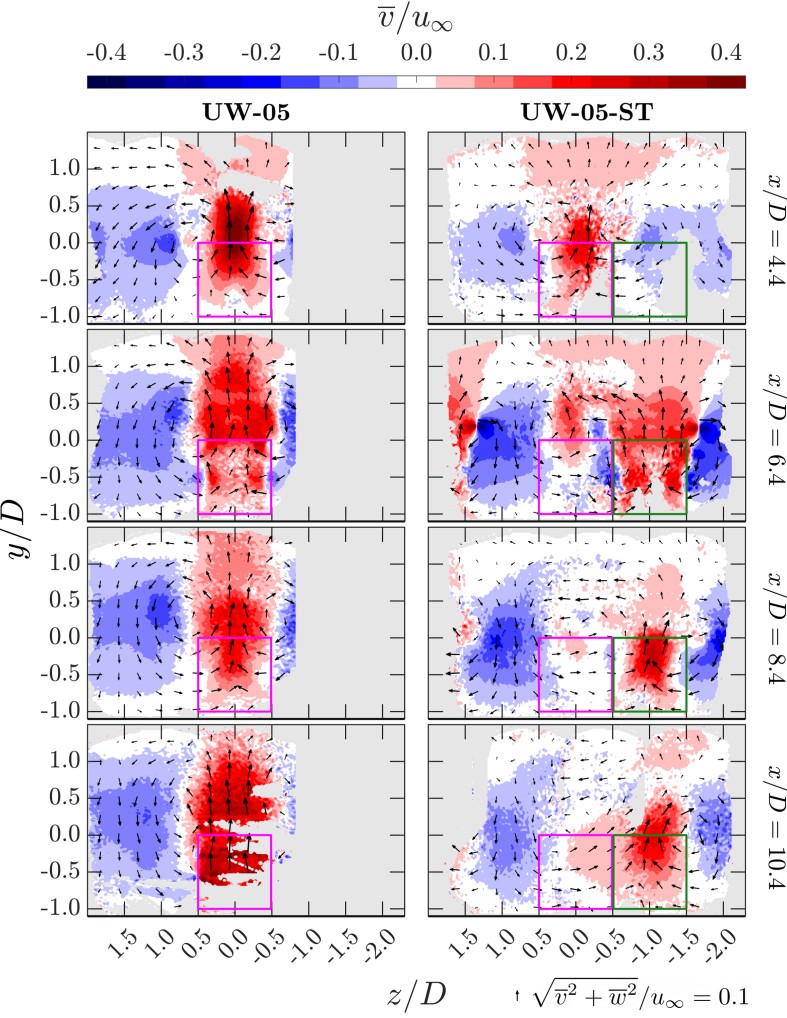

**Figure 23.** Contours of the time-averaged vertical velocity $\overline{v}$ at selected $x$-planes for the two up-washing cases. The case with the aligned layout (case **UW-05**) and the staggered layout (case **UW-05-ST**) are presented in the left and right column, respectively. The corresponding $x$-positions are indicated on the right side of each row. For the aligned case, MRSL projection areas are marked with magenta squares. For the staggered case, the projection areas of the 1st- and 2nd-row MRSLs (see Figure 7) are shown in magenta and dark green squares, respectively. In-plane velocity directions are represented by arrows scaled by their magnitude, with the reference scale provided at the bottom-right corner of the figure.





the aligned configuration draws energy from a thicker vertical region. The maximum altitude from which RGWF can entrain energy is considered critical, as it is believed to directly influence the power performance of large-scale RGWFs. Nevertheless, even at $x/D = 10.4$, which is more than $5D$ downstream from the nearest MRSL, case **UW-05-ST** still exhibits vertical flows that are stronger than $0.1u_\infty$ at $0.3D$ above the top of MRSL. This observation preliminarily confirms that the MRSL concept
remains effective, even under the tested staggered layout.

The results presented in this part preliminarily demonstrate that the concept of regenerative wind farm remains effective even with a staggered wind farm layout, supporting its robustness. However, the current study investigates only a single scenario with two rows of MRSLs. Further research involving a broader range of RGWF layouts is essential to develop a more comprehensive understanding. Such insight is particularly important for accurately estimating annual energy production (AEP) of RGWFs,
given that wind direction can vary across the full 360 degrees, giving a wide range of effective wind farm layouts. Reliable AEP prediction is critical for advancing RGWF from concept to commercial deployment, as it directly influences the calculation of the levelized cost of energy (LCoE), which is a key metric in assessing economic viability. Ultimately, financial feasibility remains one of the most dominating factors driving the development of renewable energy technologies (Lazard, 2024; McCoy et al., 2024).

## 4    Conclusions

This study experimentally investigated the aerodynamic performance of regenerative wind farms (RGWFs), a newly proposed wind farm concept. Unlike conventional wind farms that use typical wind turbines, RGWFs are composed of multi-rotor systems with lifting devices (MRSLs). These machines were specifically engineered to achieve faster wake recovery rates compared to traditional wind turbines, and this is realized through incorporating the lifting devices which are dedicated to
induce large-scale streamwise vortical structures to MRSLs (see Figure 1). Current results demonstrated that these vortical structures significantly enhanced vertical energy entrainment within RGWFs, substantially accelerating the wake recovery and improving the wind farm performance. In particular, when the MRSLs of an entire RGWFs are equipped with the lifting devices, the thrust readings of the second-row-MRSLs are more than three times than those without.

In addition to experimentally demonstrating the potential of RGWFs, the results of this study preliminarily indicated that
RGWFs perform well whether adjacent rows are aligned or staggered with the streamwise direction, reinforcing the robustness of the concept. Moreover, the current experimental findings showed strong agreement with previous numerical simulations (Li et al., 2025b, c), both showing that the wake recovery rates were accelerated due to the enhanced vertical energy entrainment driven by the vortical structures induced by the lifting devices. This cross-validation between simulations and experiments reinforces the credibility of the RGWF concept and provides a solid foundation for following research.

Despite the encouraging outcomes, several areas are recommended for further investigation. First, the current study focused on a limited set of layouts and examined only the up-washing configuration for the staggered layout. Future work should explore broader ranges of RGWF layouts, including various mis-alignment patterns, to fully characterize system performance under more realistic wind farm sites. Second, instead of using uniform inflow profiles with low turbulence level which is unrealistic



out in the field, future studies should incorporate inflow conditions that more closely resemble atmospheric boundary layers. Currently, the effects of vertical velocity shear, turbulence intensity, and thermal stability of the incoming flow on RGWF's performance remain largely underexplored and warrant systematic investigations. Lastly, since the effectiveness of RGWF has been validated through both simulations and wind tunnel experiments, the next straightforward step is to test a MRSL prototype under natural wind conditions. This test would not only validate the practicality of the concept, but would also provide valuable information on structural robustness, control system integration, and long-term energy yield in real-world environments.

In summary, the experimental results validate the aerodynamic advantages of RGWFs, demonstrating that this novel wind farm concept can substantially outperform the conventional counterpart in terms of wake recovery and land-efficiency. However, further research is required to advance RGWFs from a promising conceptual framework to a practical and cost-effective solution for harvesting renewable energy in large-scale.

*Code and data availability.* Selected measured flow field data, post-processing scripts, plotting scripts, CAD files of the experimental setup, and footage of the experiments are available in the accompanying data repository (Li et al., 2025a) (https://data.4tu.nl/private_datasets/ 6uSvxxyviKSUyqZaDS-Cspt2liwQzxrPbDppsKQrmP8).

## Appendix A: Detail descriptions about the particle tracking velocimetry system

A detailed description of the setup for the three-dimensional particle tracking velocimetry (3D-PTV) used in this work is provided in this appendix. The specifications of the software and hardware utilized are outlined in the following.

Software: Commercial software packages, DaVis 10 (by LaVision GmbH), are used for image acquisition and detecting the tracer particles.

Seeding of the tracer particles: Neutrally buoyant helium-filled soap bubbles (HFSB) with a median diameter of approximately 300 to 400 $\mu$m (Scarano et al., 2015; Faleiros et al., 2019) are used as flow tracers. The HFSB are released from an in-house developed seeding system located in the settling chamber of the wind tunnel (OJF). This system is reported to have a minimal impact on the flow. Particularly, the turbulence intensity is increased by approximately 0.5% to 0.8% (Giaquinta, 2018). The seeding system measures approximately 1,000 mm in the lateral direction and 2,000 mm in the vertical direction. However, due to the contraction at the wind tunnel exit, the lateral width of the seeded region narrows to about 570 mm as the tracers reach the region of interest. The seeding system can be slide laterally to accommodate different measurement volumes. For more detailed specifications of the seeding system, refer to Terra et al. (2024) and Bensason et al. (2025).

Illuminating source: The tracer particles are illuminated using two LED arrays that emit blue light (LED-Flashlight 300 blue, LaVision GmbH). Each array has a dimension approximately 330 mm by 110 mm. During the measurements, the LED arrays are positioned beneath the wind farm by around 1,200 mm and aligned in the streamwise direction, with their longer sides parallel to the flow. The entire FOV is observed to be adequately illuminated after the light sheet expands (see Figure 3).





Cameras and field of view: Three high-speed cameras (Photron Mini AX100) are used, each equipped with a CMOS image
sensor of 1024 px × 1024 px and a uniform pixel pitch of 20 $\mu$m. As shown in Figure 3, the three cameras are mounted on a
straight column aligned with the vertical direction ($y$-direction), with separations of approximately 700 mm. The bottom-most
camera is positioned at a height around the center of the MRSL, and the maximum angle of the cameras is about 36°. The
lateral distance ($z$-direction) between the cameras and the center of the FOV is approximately 2,150 mm. The focal lengths
of the objective lenses are 50 mm, 50 mm, and 60 mm for the bottom, middle, and top cameras, respectively. An $f\#$ of 16
is used, providing a depth of focus of around 1,000 mm. The FOV for this setup measures 830 mm, 550 mm, and 880 mm
in the streamwise, lateral, and vertical directions, corresponding to 2.8$D$, 1.8$D$, and 2.9$D$, respectively. Note that the lateral
thickness of FOV is limited by the seeding of the tracer particles. The center of FOV is located near the top of the MRSL, and
the scaling factor for this setup is approximately 0.7 mm/px.

Traverse system: The cameras and the illumination system are mounted on a traverse system (see Figure 3), enabling quick
shifts of the measurement volume. The traverse system has a range of 1,500 mm in the streamwise direction and 1,000 mm in
the lateral direction. Its precision is 1 mm. The traverse system step size in streamwise and lateral directions ($\Delta_{\text{Tra},x}$ and $\Delta_{\text{Tra},z}$)
are set to 600 mm and 375 mm, respectively. During the measurement campaign, the traverse system is manually repositioned
to cover the entire region of interest, as described in Section 2.8 and illustrated in Figure 6. Note that larger overlapping regions
between FOVs are used when the traverse system is manually repositioned.

Camera calibration: The initial geometric calibration of the cameras is performed by fitting a third-order polynomial to an
image of a calibration plate. Subsequently, a volume self-calibration is applied by taking images of flow field, reducing the
root mean square of image distortion residuals to less than 0.1 voxels. The geometric calibration is repeated in the beginning of
each day and whenever the traverse system is repositioned manually, while the volume self-calibration is frequently re-applied
to maintain the quality of the calibration.

## Appendix B: Preliminary tests for designing model of MRSL

This appendix summarizes the parameters selection process for the MRSL's aerodynamic model described in the main text,
which the model consists of a porous disk with wings attached. The selection process is based on the preliminary wind tunnel
tests. The primary objectives of these tests are to find out an appropriate airfoil profile for the MRSL's lifting devices, determine
suitable pitch angles for the wings, and provide benchmark data to guide the final design of the MRSL's model.

The preliminary tests were conducted in a low-speed wind tunnel (W-Tunnel) at the Aerodynamic Laboratories of Delft
University of Technology prior to the main experimental campaign. The wind tunnel operates as an open jet with an exit cross-
section of 600 × 600 mm$^2$. Photos of the experimental setup are shown in Figure B1. It should be noted that the floor is not
included in these preliminary tests. The coordinate axes $x$, $y$, and $z$ correspond to the (equivalent) streamwise, vertical, and
lateral directions as those in the main context.





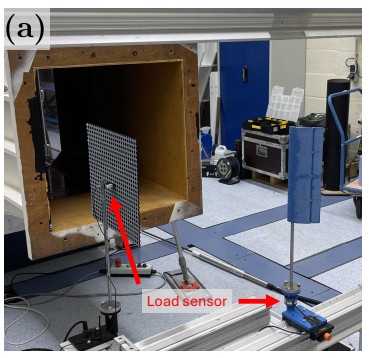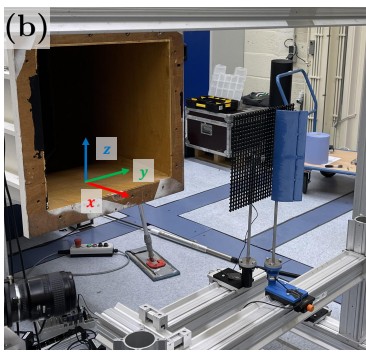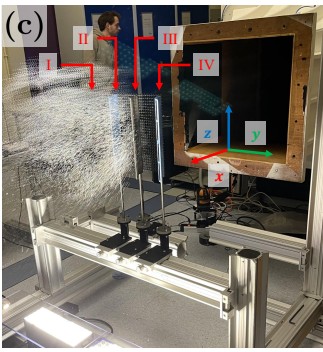

**Figure B1.** (a): Experimental setup for the preliminary tests to evaluate wing performance. The load sensors for both the wing and the porous disk are indicated by red arrows (the porous disk in this photo is positioned outside the jet). (b): Setup for evaluating the combined performance of the single wing and the disk. (c): Photograph showing the positions of the loci for the wings. The configuration shown represents the **Up-Washing** configuration, with the wing at locus **I** absent. This photo was taken during the 3D-PTV measurements.

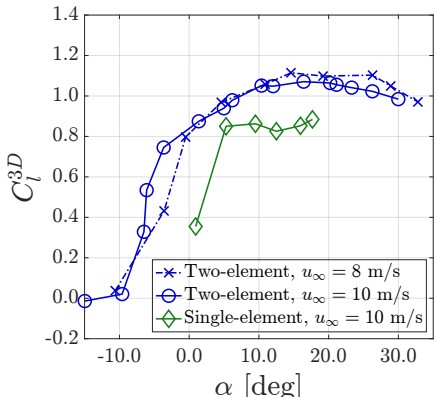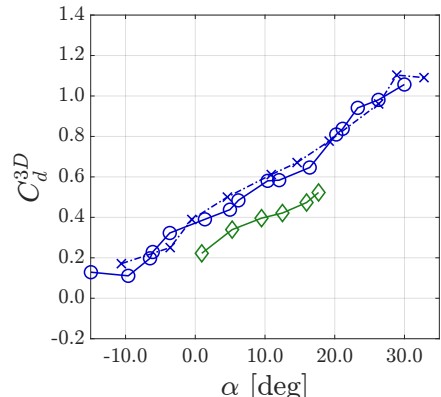

**Figure B2.** Time-averaged three-dimensional lift coefficients ($C_l^{3D}$) and drag coefficients ($C_d^{3D}$) measured in the preliminary experimental tests. The profiles of the tested airfoils are shown in Figure B3, and the experimental setup is depicted in Figure B1. The resolution of the load sensor are below $0.01 C_l^{3D}$ and $0.01 C_d^{3D}$, while the uncertainties of the angle of attack $\alpha$ are approximately $1°$.




## B1    Evaluation of the multi-element airfoil

This part benchmarks the performance of the MRSL's wings used in the final design, which their airfoil profile is categorized as multi-element airfoils. While single-element airfoils are simpler, multi-element airfoils may be preferred as they tend to offer higher maximum lift coefficients ($C_{l,\mathrm{max}}$) (Smith, 1975). For this reason, a two-element airfoil designed by Broertjes (2024) is selected as the candidate for the MRSL's model. The configuration consists of two E423 airfoils (Selig et al., 1997) and is optimized using the MSES software (Drela, 2015). The profile of the selected two-element airfoil is shown on the left side of Figure B3. Although multi-element airfoils that offer even higher $C_{l,\mathrm{max}}$ exist, this particular profile is chosen for its relatively simple structure (facilitating manufacturing), high thickness-to-chord ratio (aiding assembly and installation), and moderate camber (reducing flow blockage). Additionally, for comparison, a single-element S1223 airfoil (shown on the right of Figure B3) (Selig et al., 1995) is also tested to verify whether the selected two-element airfoil provides superior lift performance.

For the tested wings, the span and chord are set to 300 mm and 100 mm, respectively, matching the dimensions of the final MRSL design. Their performance is evaluated using three-dimensional lift and drag coefficients ($C_l^{3D}$ and $C_d^{3D}$). The forces are measured using a six-axis force-torque sensor (F/T Sensor: mini40 with SI-40-2 calibration, ATI Industrial Automation), with an precision of 0.01 N. The sampling rate is set to 500 Hz, and the measurements are conducted over a duration of 20 s. The wings and sensors are mounted on a frame using a combination of metallic (LINOS X95 System) and 3D-printed components, as shown in FigureB1.

$C_l^{3D}$ and $C_d^{3D}$ for the wings with the two airfoil profiles are plotted as functions of the angle of attack ($\alpha$) in Figure B2. The tests are conducted at reference velocities of 8 m/s and 10 m/s for the wings with the two-element airfoil, and at 10 m/s for those with the single-element airfoil. Angles of attack are measured before and after testing using a camera and image processing software (ImageJ (Schneider et al., 2012)), with measurement uncertainties estimated to be approximately $1°$. The locations of the pitching axes for both airfoils are shown in Figure B3.

Based on the $C_l^{3D}$ results shown in Figure B2, the maximum $C_l^{3D}$ achieved by the wings with the two-element airfoil profile is approximately 25% higher than that of the single-element airfoil. According to classic lifting-line theory (Anderson, 2011), increased lift is associated with stronger tip vortices, which in turn enhance the induced vertical velocity in the wake (commonly referred to as downwash). Note that stronger induced vertical velocity is considered beneficial for the performance of MRSLs (Li et al., 2025c). Therefore, despite the higher $C_d^{3D}$ exhibited by the two-element airfoil, it is selected for the final MRSL design due to its superior lift characteristics, which provide aerodynamic advantages that could accelerate the downstream flow recovery.

## B2    Selecting the pitch angles of the MRSL

The performance of the wing with the two-element airfoil profile, characterized by $C_l^{3D}$ and $C_d^{3D}$, is initially evaluated under freestream conditions. However, since the wings in the MRSL's model are expected to undergo significant aerodynamic interaction with the porous disk, it is necessary to measure the loads on the wings and the porous disk when they are assembled





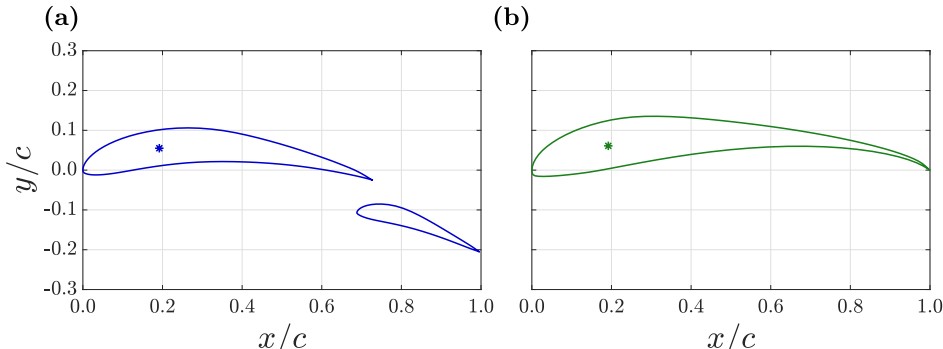

**Figure B3.** (a): The XY-plot depicts the two-element airfoil used in this study, designed by Broertjes (2024) using the MSES optimization tool (Drela, 2015). This configuration consists of two E423 airfoils (Selig et al., 1997). (b): The XY-plot shows the single-element airfoil, S1223 (Selig et al., 1995), used as a reference to evaluate the performance of the two-element airfoil employed in this work. The pitching axes for both airfoils are indicated with $\star$ and are located at $(x/c = 0.192, y/c = 0.055)$ for the two-element airfoil and $(x/c = 0.192, y/c = 0.061)$ for the single-element airfoil.

together. This approach enables the selection of appropriate pitch angles $\theta_p$ for the wings based on measured loads, with the objective of maximizing the wings' vertical ($y$-directional) forces.

To independently measure the loads on the wings and the porous disk, it is essential to eliminate any solid contact between the two components. This is accomplished using the setup shown in Figure B1(b). Additionally, because the relative positions of the wings and the disk are expected to influence their aerodynamic interaction, load measurements are conducted at every mounting position of the wings.

    Prior to the experiment, it has already been determined that MRSL is going to be equipped with three wings and both up-
washing and down-washing configurations is going to be investigated. Since the preliminary tests do not include the floor, some wing loci can be used for both configurations, reducing the number of required test positions to four. The positions of these loci are illustrated in Figure B1(c). Specifically, for the up-washing configuration, the wings are placed at loci **IV**, **III**, and **II** as the top, middle, and bottom wings, respectively. For the down-washing configuration, the wings are positioned at loci **I**, **II**, and **III** as the top, middle, and bottom wings, respectively. The positions of these four loci are located at $y = -0.05D, 0.32D,$
$0.68D$, and $1.05D$, with the origin placed at the edge of the disk on the negative $y$ side.

    The time-averaged three-dimensional wing's loads measured in the presence of the porous disk are displayed in Figures B4(a) and B4(b). Note that only one wing is installed at a time and the porosity of the disk is $60\%$. Here, $C_y$ and $C_x$ represent the vertical and streamwise force coefficients, respectively, as defined in Equation (B1), with $u_\infty = 8.0$ m/s. $f_y^W$ and $f_x^W$ are the streamwise and vertical forces exerted by the wing. Notation of $C_l^{3D}$ and $C_d^{3D}$ are not used here, as the presence of the disk
complicates and alters the inflow conditions, including the direction of the inflow. The results show that $C_y$ for loci **I**, **II**, and **III** generally under-perform compared to cases without the disk. This is expected, as the porous disk slows down the flow, reducing the forces, a trend also observed in $C_x$. Interestingly, $C_y$ for locus **IV** significantly outperforms the case without the



disk. This can be attributed to the fact that the wing at locus **IV** is not shadowed by the porous disk. Additionally, the blockage effect of the disk accelerates the flow and alters its direction, which further enhances $C_y$ for locus **IV** compared to conditions

without the disk.

$$C_y \triangleq \frac{|f_y^W|}{0.5\rho u_\infty^2 cS}, \qquad C_x \triangleq \frac{|f_x^W|}{0.5\rho u_\infty^2 cS} \tag{B1}$$

Regarding the maximum vertical force (maximum $C_y$) measured at different loci, it is observed that the maximum $C_y$ for loci **I**, **II**, and **III** occurs around $\theta_p = 15°$, while for locus **IV**, it occurs at $\theta_p = 25°$. However, a significant drop in $C_y$ is seen at locus **IV** when $\theta_p > 25°$, prompting a more conservative choice. Additionally, visualizing the flow field with smoke reveals

severe flow separation at locus **I** when $\theta_p$ is set beyond $10°$, which is only alleviated when $\theta_p \leq 5°$. Based on these findings, the pitch angles for the final design are set as follows. For the up-washing configuration, $\theta_p$ for top, middle, and bottom wings are set at $-15°$, $-15°$, and $-15°$, respectively. As for the down-washing configuration, $\theta_p$ are set to $5°$, $15°$, and $15°$ for top, middle, and bottom wings, respectively. The values of $\theta_p$ given here are those used in the final design described in Section 2.2.

The thrust coefficient $C_T$ of the porous disk, which is measured simultaneously with wing's $C_y$ and $C_x$, is presented in

Figure B4(c). The load sensor used for the disk is a unidirectional strain gauge (KD24s 10N, ME-Me$\beta$systeme) capable of measuring tensile and compressive forces with a precision of $0.1$ N. The sampling rate is set to $1,000$ Hz, and the sampling duration is $20$ s. The plot shows that the load on the disk is generally higher when the wing is positioned at loci **III** or **IV** compared to loci **I** or **II**. This result aligns with expectations, as the bound circulation of the wing alters the flow field around the disk. When the wing is situated at loci **III** or **IV**, its circulation system tends to accelerate the flow over the disk, whereas

the circulation of the wing at loci **I** or **II** slows the flow down.

Interestingly, despite the additional blockage and drag introduced by the wing, the presence of a wing at loci **III** or **IV** can result in higher loads on the disk compared to a stand-alone disk. This phenomenon is also observed in measurements of the thrust force for the up-washing configuration in Section 3.1. These observations highlight the significant aerodynamic interactions between the wings and the porous disk of the MRSL's model.

**B3  Concluding remarks for preliminary experimental tests**

The preliminary experiments provide valuable insights about the aerodynamic behaviors of the MRSL model and is critical in informing the final design used in the main study for farm-scale testing. Notably, three-dimensional PTV measurements are also conducted during these tests. While the PTV results contribute to the design decisions, they are not included in this appendix, as many of the key findings have already been covered in the main text. However, several interesting and potentially

important aspects of the MRSL model remain unexplored, such as the wing-wing aerodynamic interactions within an MRSL. These topics are left for future investigations.



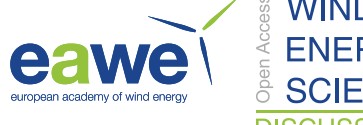

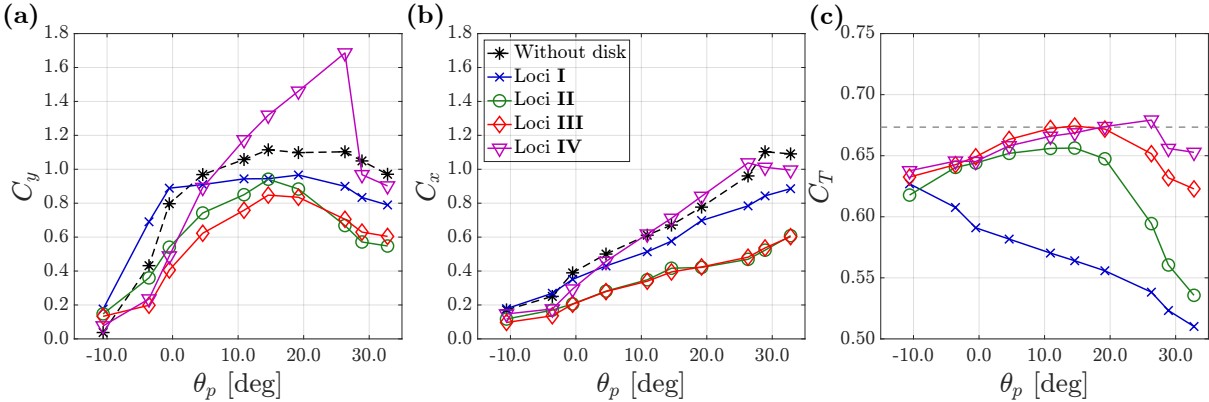

**Figure B4.** (a) and (b): Time-averaged three-dimensional wing's loads measured in the presence of the porous disk with porosity being $60\%$. Note that only a single wing (two-element airfoil) is placed during the measurements, as depicted in the middle panel of Figure B4. Wing loads are measured at different positions, with the loci positions illustrated on the right of Figure B4. $C_y$ and $C_x$ denote the vertical and streamwise force coefficients, respectively, as defined in Equation (B1). The uncertainties of $C_x$ and $C_y$ due to the load sensor are below $0.01$, while the uncertainties in the pitch angle $\theta_p$ are approximately $1°$. (c): The corresponding thrust coefficient $C_T$ measured simultaneously with $C_y$ and $C_x$. The gray dashed line indicates the $C_T$ level for the disk without wings. The uncertainties in $C_T$ due to the load sensor are below $0.03$.

## Appendix C: Algorithm for stitching and scaling the measured volumes

This appendix provides a detailed description of the stitching algorithm used to stitch the volumes measured with 3D-PTV. An overview of the stitching process is presented in the diagrams in Figure C1, while the algorithm itself is described in detail from Equation (C1) to (C4). The scripts for executing stitching along with an example data set are provided with the accompanying data repository (Li et al., 2025a).

During the stitching process, volumes are stitched recursively, as illustrated in Figures C1(b) and C1(c). It is important to note that the algorithm handles only one direction at a time, requiring the stitching sequence to be designed in advance.

In Figure C1 and Equations (C1) to (C4), $\boldsymbol{x}$ represents the position vector of the point of interest, while $\boldsymbol{\zeta}$ denotes the position vector of the stitching reference point within volume *Main*, which the vector is used to determine the blending weightings. Throughout this appendix, the suffix $i$ indicates the stitching direction, which in this work corresponds to $x$ (streamwise direction) or $z$ (lateral direction). The variable $\delta_i$ is the distance between $\boldsymbol{x}$ and $\boldsymbol{\zeta}$ along the $i$th-direction.

The blending weightings $f_{\mathrm{blend},i}$ and $f_{\mathrm{antiblend},i}$ are defined in Equation (C2) and are described using a hyperbolic tangent function within the blending zone. The constant $\Lambda_{\mathrm{blend}}$ controls the rate of change of the blending weights, and in this work, $\Lambda_{\mathrm{blend}} = 3$ is used (note that $\tanh(3) > 0.995$). These blending weightings help alleviate the potential spurious jumps in flow quantities due to stitching.

When stitching two volumes, the blending depth is defined as $2\,l_{\mathrm{blend},i}$. In this work, $l_{\mathrm{blend},x}$ and $l_{\mathrm{blend},z}$ are both set to 75 mm. This choice is based on the traverse system step size ($\Delta_{\mathrm{Tra},x} = 600$ mm and $\Delta_{\mathrm{Tra},z} = 375$ mm) and the field of view dimensions



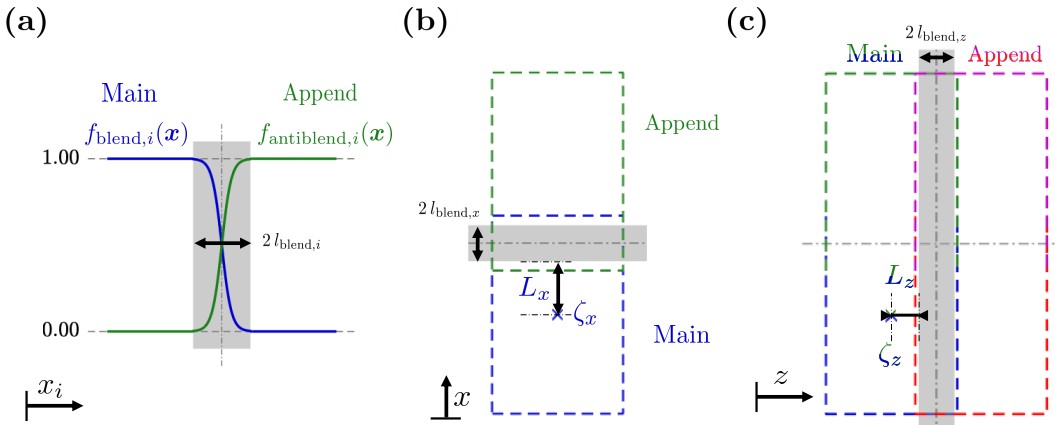

**Figure C1.** Schematic diagrams illustrating the stitching of measured volumes. The dashed lines in different colors represent the fields of views from different measurements. In the diagrams, volume *Append* is stitched onto volume *Main*, with the shaded areas indicating the blending regions. (a) Blending weights for volume *Main* and anti-blending weights for volume *Append*. (b) Stitching volume *Append* onto volume *Main* in the $x$-direction. (c) Stitching another volume *Append* onto the existing volume in the $z$-direction. Note that both *Main* and *Append* can represent volumes that have already been stitched together with several sub-volumes.

(FOV$_x$ = 830 mm and FOV$_z$ = 550 mm), ensuring that the entire blending zone falls within the overlapping region of the two measurements. Specifically, the condition $\Delta_{\mathrm{Tra},i} + 2\,l_{\mathrm{blend},i} < \mathrm{FOV}_i$ is satisfied for both the $x$ and $z$ directions. This is visualized in Figures C1(b) and C1(c). Note that $L_i \equiv (\Delta_{\mathrm{Tra},i} - 2l_{\mathrm{blend},i})/2$.

$$\boldsymbol{\delta}(\boldsymbol{x}) = \boldsymbol{x} - \boldsymbol{\zeta}, \qquad \delta_i = x_i - \zeta_i \tag{C1}$$

$$f_{\mathrm{blend},i}(\boldsymbol{x}) = \begin{cases} 1.00 & \text{if } \delta_i(\boldsymbol{x}) < L_i \\ 0.00 & \text{if } \delta_i(\boldsymbol{x}) > L_i + 2\,l_{\mathrm{blend},i} \\ 0.5\left(\tanh\left[\Lambda_{\mathrm{blend}}\left(\frac{L_i - \delta_i}{l_{\mathrm{blend},i}} + 1\right)\right] + 1\right) & \text{if } L_i \leq \delta_i(\boldsymbol{x}) \leq L_i + 2\,l_{\mathrm{blend},i} \end{cases}, \quad f_{\mathrm{antiblend},i}(\boldsymbol{x}) = 1 - f_{\mathrm{blend},i}(\boldsymbol{x}) \tag{C2}$$

Before initiating the stitching process, the binned data is first interpolated onto a global grid shared by all volumes. The grid size of the global grid is comparable to the binning size. This step significantly facilitates the algorithm since the voxel sizes of different measurements may vary slightly due to re-calibration between measurements. After projecting the binned data onto the global grid, a layer equivalent to two grid sizes is trimmed from the edges to reduce noise before further processing.

An important consideration in the stitching process is that the volumes being stitched together are obtained from different measurements. Due to limitations of the wind tunnel, the jet wind speed is not perfectly consistent over the weeks-long experimental campaign, fluctuating between 7.2 and 7.4 m/s. To prevent spurious discontinuities, slight scaling of the flow properties





is done during stitching. The scaling factor, denoted as $\Pi$, is calculated through Equation (C3) based on the time averaged streamwise velocity $\overline{u}$ measured in the blending zone overlapped by the two volumes.

$$\Pi = \left[ \sum_{\substack{\boldsymbol{x} \in \text{blending zone} \\ \min(N_{\text{Main}}, N_{\text{Append}}) \geq \Theta_N \\ y \geq \Theta_y}} \overline{u}_{\text{Main}} \min\left(N_{\text{Main}}, N_{\text{Append}}\right) \right] \Bigg/ \left[ \sum_{\substack{\boldsymbol{x} \in \text{blending zone} \\ \min(N_{\text{Main}}, N_{\text{Append}}) \geq \Theta_N \\ y \geq \Theta_y}} \overline{u}_{\text{Append}} \min\left(N_{\text{Main}}, N_{\text{Append}}\right) \right] \quad \text{(C3)}$$

To minimize noise when calculating the values of $\Pi$, only the upper half of the blending region ($y \geq \Theta_y$, $\Theta_y/D = 0.2$) is considered, as this region is generally less affected by the MRSL and exhibits lower turbulence. Additionally, a particle count threshold of $\Theta_N = 10$ is applied to ensure that $\Pi$ is less influenced by noise (data is neglected if $N < \Theta_N$).

Finally, with $f_{\text{blend},i}$, $f_{\text{antiblend},i}$, and $\Pi$, the stitching process can be performed. The stitching of a property $B$ is described mathematically in Equation (C4). Here, $n_B$ represents the exponent of the scaling factor $\Pi$, and its value depends on the nature of $B$. This is necessary because $\Pi$ is determined based on the streamwise velocity component $u$, regardless of what $B$ represents. For velocity components ($u$, $v$, or $w$) or vorticity components ($\omega_x$, $\omega_y$, or $\omega_z$), $n_B = 1$. For properties such as turbulent kinetic energy or components of Reynolds stress, $n_B = 2$. Additionally, if $B$ represents particle number $N$, $n_B$ is set to 0, as no scaling is required in this case.

$$B_{\text{stitched}}(\boldsymbol{x}) = \begin{cases} B_{\text{Main}} & \text{if } \delta_i(\boldsymbol{x}) < L_i \\ \Pi^{n_B} B_{\text{Append}} & \text{if } \delta_i(\boldsymbol{x}) > L_i + 2\,l_{\text{blend},i} \\ B_{\text{Main}} & \text{if } L_i \leq \delta_i(\boldsymbol{x}) \leq L_i + 2\,l_{\text{blend},i} \text{ and } N_{\text{Main}} \geq \Theta_N \text{ and } N_{\text{Append}} < \Theta_N \\ \Pi^{n_B} B_{\text{Append}} & \text{if } L_i \leq \delta_i(\boldsymbol{x}) \leq L_i + 2\,l_{\text{blend},i} \text{ and } N_{\text{Append}} \geq \Theta_N \text{ and } N_{\text{Main}} < \Theta_N \\ \frac{B_{\text{Main}} N_{\text{Main}} f_{\text{blend, i}} + \Pi^{n_B} B_{\text{Append}} N_{\text{Append}} f_{\text{antiblend, i}}}{N_{\text{Main}} f_{\text{blend, i}} + N_{\text{Append}} f_{\text{antiblend, i}}} & \text{if } L_i \leq \delta_i(\boldsymbol{x}) \leq L_i + 2\,l_{\text{blend},i} \text{ and } N_{\text{Main}} \geq \Theta_N \text{ and } N_{\text{Append}} \geq \Theta_N \\ \text{NaN (set to 0 if } B \text{ is } N) & \text{else} \end{cases}$$

$$\text{(C4)}$$

After recursively stitching the measured volumes, the entire field of interest is obtained as a single entity, resulting in a single stitched volume for each case. However, to facilitate comparisons between cases with different configurations of the regenerative wind farm, a final scaling step is required to account for slight variations in inflow velocity between cases (due to wind tunnel). For this purpose, the property $B$ of the stitched volumes is multiplied by $\Pi_{\text{Global}}^{n_B}$, where $\Pi_{\text{Global}}$ is determined in the inflow zone of the regenerative wind farm, as described in Equation (C5). For the cases with the aligned layout, the inflow zone is defined within the region $-0.95 < x/D < -0.60$, $0.53 < y/D < 0.88$, and $0.84 < z/D < 1.20$, and the reference inflow velocity is given as $u_\infty = 7.3$ m/s.





$$\Pi_{\text{Global}} = u_\infty \left/ \left[ \left( \sum_{\substack{\boldsymbol{x} \in \text{inflow zone} \\ N_{\text{stitched}} \geq \Theta_N}} \overline{u}_{\text{stitched}} N_{\text{stitched}} \right) \left/ \left( \sum_{\substack{\boldsymbol{x} \in \text{inflow zone} \\ N_{\text{stitched}} \geq \Theta_N}} N_{\text{stitched}} \right) \right. \right] \right. \tag{C5}$$

*Author contributions.* YL: prepared and performed experiment, post-processing, formal analysis, writing. MF: prepared and performed experiment, post-processing, formal analysis, writing. BD: prepared and performed experiment. WY: supervision and technical review. AS: supervision and technical review. CF: conceptualization, performed experiment, supervision, and technical review.

*Competing interests.* The contact author has declared that neither they nor their co-authors have any competing interests.

*Acknowledgements.* The authors would like to thank Haoyuan Sun, Shantanu Purohit, Manuel Ratz, and Jayant Mulay for their assistance during the experiments, Ed Roessen and Stefan Bernardy for their technical support, and Valentin Le Bailly de Tilleghem together with the other group members of DSE-06 (Carraro et al., 2024) for providing the computer render presented in Figure 1.





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
