# Peer review of "Experimental Study of Regenerative Wind Farms Featuring Enhanced Vertical Energy Entrainment"

_Wind Energy Science, 2025_

## Author Comment (AC1)

**Reply to Reviewers**

Manuscript ID WES-2025-156

**General overview**

We sincerely appreciate the two reviewers for their time and dedication in reviewing our work. It is our pleasure that both of the reviewers find our article interesting and affirms the quality of the presentation. In this document, we respond to the comments given in details. We believe given the inputs provided by the reviewers have further enhance the quality and the completeness of the manuscript.

**Reply to comments from Reviewer #1**

**General review:**

1. There is a comparison with CFD results from a different published study, which however only has a 'broadly comparable' set up. This means the geometry is different and the Re number is orders of magnitude different. I understand there is good qualitative agreement between the experiments and the CFD results, but the relevant section seems to weaken the work rather than strengthen it. If the comparison with the published paper is significantly reduced in size and placed in the right context, that would help the present manuscript.

**Reply:**

We thank the reviewer for this thoughtful insight. We acknowledge that the comparison with our previous CFD studies does not follow the conventional sequence that is usually done in academia, where experiments are typically conducted first and then used to validate the following numerical modeling. In our case, simulations are carried out prior to the experiments, as they are considerably faster and more economical than wind tunnel testing, and they have provided the first indication of the strong potential of RGWFs. While the computational approaches we use are verificated and considered high fidelity, the lack of experimental evidence has been a major shortcoming in our earlier work.

For this reason, we believe it is important to use the present experimental data to reinforce the findings of our previous numerical studies. The intention of the comparison is therefore not to claim quantitative agreement, but to demonstrate qualitative consistency across independent methodologies, which we consider a critical step in validating the overall concept.

That said, we agree that the objective of this section is not clearly articulated in the original manuscript, which may have led to the reviewer's concern. The section is included not because we lack confidence in the experimental results, but rather to demonstrate that two independent methodologies consistently support each other. To clarify this point, we have revised the section to better frame the comparison in the appropriate context and make the purpose explicit.

For example, we have added "... The primary aim of this part is to demonstrate that the predictions from these prior simulations align well with the current experimental data, thereby reinforcing the credibility of the conclusions drawn based on those numerical outcomes. ..." to the introductory paragraph in Section 3.6.2 of the revised manuscript.

Moreover, several other minor but non-trivial insights into the aerodynamics of MRSLs can be drawn by comparing the experimental results with the numerical predictions, particularly given the discrepancies in the key parameters (e.g., Re\_D). For instance, we have long hypothesized that the key parameter governing the wake aerodynamics of MRSLs is the lift-to-thrust ratio. The present comparison provides partial evidence supporting this hypothesis. We have now made this point explicit in Section 3.6.2. "Besides, the comparison also provides some other insights. For instance, that the general wake aerodynamics of MRSLs appears to be mainly governed by the thrust coefficient and the lift-to-thrust ratio." and "These observations not only highlight the discrepancies between the datasets but also may provide insight for future MRSL optimization, such as fine-tuning wing placement and pitch settings to achieve a more effective distribution of lift across the MRSL's height.".

2. The discussion on the staggered case can be deeper.

**Reply:**

We thank the reviewer for pointing out this shortcoming. In the revised manuscript, we add Section 3.7.2, which provides a deeper discussion of the staggered case (UW-ST-05) by analyzing the area-averaged available power.

3. Given the beautiful dataset, the analysis of flow features could be more substantial

**Reply:**

We thank the reviewer for this valuable suggestion. We agree that our dataset offers plenty of opportunities for deeper exploration of fundamental fluid mechanics aspects. For example, (1) the vortex dynamics of the three trailing-vortex pairs observed in Figure 17, including their mutual interactions and possible merging, vortex breakdown, and their interaction with the floor, (2) the behavior of these vortices within the highly sheared layer appear in the perimeter of the porous disk wake, including whether the shear layer influences their diffusion or meandering. These questions are indeed interesting in their own right and may also have practical importance for understanding the detailed flow physics of RGWFs.

However, the main objective of the present manuscript is to provide experimental validation of the regenerative wind farm concept in mitigating wake losses. As a result, some flow-field quantities that are important for detailed fluid-dynamics analysis, particularly higher-order statistics, were not captured with high quality, preventing precise quantitative analysis straight forward. For this reason, we choose not to pursue such in-depth analyses here and instead leave them for future studies, which are already being planned. That said, we expand the presentation and the discussion of the vorticity fields for cases UW-05 and DW-05, which is now presented in Appendix D of the revised manuscript.

**Minor comments appear in the attachment:**

Short note: The minor grammatical errors and typos are not detailed in this document, please refer to "author track changes". Also, if not mentioned otherwise, all the line-numbers mentioned in this document refer to the version of initial submission.

**Abstract**

1. The reviewer suggests replacing the term "airfoils" with "wings" in line 6, as airfoils implies a focus on two-dimensional aerodynamics, whereas our study addresses three-dimensional effects.

**Reply:**

We thank the reviewer for this careful observation. We agree that "wings" is more appropriate than "airfoils" in the present context.

**Introduction**

1. In line 24, the reviewer raised that the reference of farm efficiency drop is not completely clear. "... the losses caused by turbine–turbine wake interactions within wind farms, which can reduce overall farm efficiency by 10 to 25%."

**Reply:**

We thank the reviewer for raising this point. However, we believe that the original sentence is sufficiently clear for readers within the community of Wind Energy Science, and providing additional detail may be redundant. A more literal rephrasing of the sentence would be, "Farm efficiency is reduced by 10 to 25% due to losses caused by turbine–turbine wake interactions." This is essentially equivalent to stating that, in the absence of wake interactions (i.e., if all turbines were exposed to free-stream wind), a 10 to 25% reduction in farm efficiency could be

avoided. Since the meaning is already conveyed and the sentence fits well within the overall flow of the introduction, we have chosen not to modify the text.

2. In line 33, the reviewer would like us to clarify the "specific implementation approach" mentioned.

**Reply:**

We thank the reviewer for this helpful suggestion. To improve clarity, we revise the text to: "... regardless of the specific implementation approach. That is, whether the energy-harvesting element is a conventional single-rotor horizontal-axis wind turbine or a system with multiple vertical-axis wind turbines, the principle remains effective as long as large-scale lift-generating elements are incorporated.". We believe this change enhances the clarity of the statement.

**Methodology**

1. In line 73, the reviewer points out that we did not clearly mentioned how the supporting structures of MRSL, such as the scaffolding, are treated.

**Reply:**

We thank the reviewer for pointing out this oversight. In our study, we do not intend to capture the aerodynamic effects of the supporting structures. This is now clarified at the end of Section 2.2 with the following text: "Note that the steel rod is not intended to replicate the aerodynamic effects of a wind turbine tower, which its purpose is purely for holding the model. In fact, the supporting structures of the multi-rotor system (e.g., scaffolding) are not represented in the present aerodynamic model, and their aerodynamic impacts are therefore not considered.".

2. In line 92, the reviewer would like us to clarify the "thrust" mentioned, whether it is referring to the whole structure (porous square and 3x wings) or just the wings.

**Reply:**

The "thrust" mentioned refers specifically to the force exerted by the porous disk only. We now make this explicit in the revised text.

3. In line 121, the reviewer would like us to clarify the "thrust, drag, lift" mentioned.

**Reply:**

We now explicitly clarify the force definitions. That is, thrust refers to the streamwise force exerted by the porous disk, drag refers to the streamwise force exerted by the wings, and lift refers to the vertical force exerted by the wings. The text has been revised to: "To quantify the forces exerted by MRSLs, both streamwise (thrust of the porous disk and drag of the wings) and vertical (lift of the wings) forces are measured for the MRSLs located in the mid-column." Together with the subsequent explanation in this subsection, we believe the definitions of these forces are now clear.

4. The reviewer pointed out Section 2.8 may could be omitted for brevity.

**Reply:**

We understand the reviewer's point that Section 2.8 may not be essential if the reader's interest is purely in the results and conclusions. However, we would like to retain this section, as we believe it provides important information and valuable guidance for other researchers who may wish to reproduce similar experiments. Large-scale 3D-PTV experiments are extremely rare in the literature, and given the quality of the dataset obtained within a relatively short period, we consider a detailed description of the procedure highly beneficial. Many aspects of the workflow are not straightforward before conducting the experiments, and presenting them in detail also helps explain certain limitations of the present measurement campaign (e.g., slight discontinuities in the velocity fields when stitching data collected on different days). For these reasons, we believe the section is informative and useful, particularly for experimental aerodynamicists, and therefore we have chosen to keep it.

5. In line 220, the reviewer has doubt of the word choosing of "recipe".

**Reply:**

We acknowledge that "recipe" may not be the most conventional choice of wording in this context. However, we believe that it introduces very little ambiguity and helps readers grasp the meaning quickly. In addition, we feel that this expression adds a touch of liveliness to an otherwise fact-based manuscript, which may improve readability without detracting from scientific rigor.

**Results and discussions**

1. In the first paragraph of this section, the reviewer pointed out already highlighting the results in the introductory paragraph may not be very appropriate.

**Reply:**

We thank the reviewer for this suggestion. We have revised the introductory paragraph of the section by removing the results and discussions, keeping it focused only on setting the context.

2. In Section 3.1, the reviewer pointed out the context of the first two paragraphs is more suitable to appear in the Section Methodology.

**Reply:**

We thank the reviewer for this helpful suggestion. Upon re-reading the manuscript, we deeply agree that the content of the first two paragraphs of Section 3.1 fits better in Methodology. We have therefore moved this text to Section 2.4 in the revised manuscript.

3. *In line 247, the reviewer asked about how the density is decided.*

**Reply:**

We thank the reviewer for pointing out this missing information. We now clarify in Section 2.4 that the density value is determined from the wind tunnel readings, based on pressure and temperature. We also appreciate that this comment motivated us to re-check the density value, which we have overlooked in our initial submission. That is, the density should be 1.205 kg/m³, but was mistakenly written as 1.225 kg/m³. We note that all force coefficients in the manuscript are calculated using the correct density in the analysis script, so no data adjustments are required. If using 1.225 kg/m³, the calculated thrust coefficient should be 0.70 instead of 0.72.

4. In the caption of Figure 11, the reviewer pointed out the standard deviations of  $D^W$  are not presented.

**Reply:**

We thank the reviewer for this comment. The standard deviations of D^W are not presented because they are obtained by subtracting the averages of two time-series datasets that are not measured concurrently, as explained in Section 2.4. Although methods exist to estimate the standard deviations for such data, we do not perform this analysis here, as the additional information would be very limited.

5. In Figure 13, the reviewer pointed out it will be worthwhile to resolve the vortex dynamics, such as merging, if the resolution of the data allows.

**Reply:**

We are very glad that the reviewer identified many interesting flow features around the MRSL, just as we did. Indeed, we agree that resolving the vortex dynamics, such as the merging process, would be valuable. However, as we would like to keep our focus on experimentally validating the concept of RGWF fine-scale vortex dynamics are not deeply studied. Instead, we focus on the larger-scale flow features relevant to wake recovery. However, motivated by the reviewer, we have now provided a more detailed set of vorticity plots for cases UW-05 and DW-05 in Appendix D, where the evolution and diffusion of the tip vortices are better illustrated. As for detailed MRSL's vortex dynamics, including merging, in future studies, is planned to be investigated in the future studies.

6. In Figure 13, the reviewer would like to confirm if negative time averaged streamwise velocity appeared.

**Reply:**

We thank the reviewer for raising this relevant concern. After re-checking our data, we confirm that no regions with negative time-averaged streamwise velocity are observed. However, we do identify small patches with velocities very close to zero around the second-row MRSL in case UW-05. This has now been explicitly described in the caption of Figure 13.

7. In line 335, the reviewer understand that the slight discontinuity due to combining data from different measurement sessions do not significantly impacts the analysis and conclusion made in this manuscript, but the reviewer is wondering if we could quantified the uncertainty, as he suggest this information would be valuable if one would like to reuse our data. For example, to validate one's CFD results.

**Reply:**

We thank the reviewer for raising this important point and for highlighting that our dataset may add value to the community as a benchmark for CFD validation. Based on the data presented in Figure 18 (available power), we find that the discontinuity in mean u can exceed 10% of u\_infty (now explicitly mentioned in Section 3.5 of the revised manuscript), which is considerably larger than the 95% confidence interval shown in Figure 9. Unfortunately, we cannot provide a single, concrete recommendation for the uncertainty level that future studies should adopt when reusing our data, as this depends on the specific application. For example, the uncertainty in absolute values of u may be relatively low in the upstream-most traverse positions but higher downstream, whereas for spatial derivatives the uncertainty distribution may differ and could be less severe. We therefore leave it to future users to judge the uncertainty according to their purpose and needs.

That said, we have disclosed all data-processing details, the full experimental procedure, and the processing scripts along with the measurement data, which we believe provide sufficient information for others to make their own informed judgment. To guide readers more clearly, we have added the following sentence in this paragraph: "A more quantified presentation can be found in the later section (see Section 3.5)." In addition, the quantified value mentioned above is now documented in the last paragraph of Section 3.5, providing a more precise reference for the reader (exceeding 10% of u infty).

8. In line 340, the reviewer points out it is uncommon to use numerical data to enhance the credibility of experimental data. He or she recommend us to modify the statement.

**Reply:**

We thank the reviewer for pointing out this peculiarity. The main purpose of the statement is to acknowledge that our experimental data have certain limitations, and that cross-checking them with numerical data helps reinforce our overall objective, namely, to demonstrate the strong potential of RGWF in mitigating losses caused by turbine—turbine wake interactions. We agree that the original wording may not have clearly conveyed this intent and could be interpreted as suggesting that numerical data are used to validate experimental data. To address this, we have revised the sentence from "... not only validates the numerical simulations using experimental data but also enhances the credibility of the present experimental findings ..." to

"... not only validates the numerical simulations using experimental data but also consolidates the present experimental findings despite data imperfections.".

9. *In Figure 15, the reviewer is wondering whether we presented all the arrows.*

**Reply:**

We do remove two arrows at the bottom ( $y/D \sim = -1.1$ ) for DW-05, not due to data quality but because they extended outside the figure bounding box and disrupted the figure layout. Apart from this minor adjustment, all arrows are presented and preserved. We consider this manipulation trivial enough that it does not require explicit mention. We also note that noise is minimal due to the data trimming described in Section 2.6 and Appendix C.

10. In line 365, the reviewer is not really certain about why we mention that further experimental data focusing on the impacts of turbulence on lifting devices is crucial for future development of MRSL. He or she is wondering about if we are concerning turbulent spectrum.

**Reply:**

We would like to emphasize that the aerodynamic response of MRSLs to turbulence remains underexplored, and further experimental data are needed before one can fully assess their performance in real-world turbulent environments. This point is particularly important because previous studies on wind turbine aerodynamics have shown that turbulence can substantially affect the persistence of structured vortical flows, of which the streamwise vortical structures induced by the lifting devices fall in to this category. Note that this is clearly stated in the first paragraph of Section 3.3.

At this stage, we focus on turbulence intensity rather than the turbulent spectrum. To avoid confusion and to keep the discussion focused, we choose not to mention the turbulent spectrum in the manuscript. In addition, we find the sentence "Such data would be critical for translating the proposed concept into real-world applications." to be redundant, and therefore we have removed it.

11. In line 370, the reviewer suggests to overlay the TI contours with vorticity isolines to consolidate the claim about the vortex centers have higher measured TI.

**Reply:**

We understand the reviewer's suggestion and agree that overlaying TI contours with vorticity isolines could provide more straightforward presentation. However, since the primary focus of this manuscript is not on the detailed vortex dynamics of MRSL, we believe that pinpointing this claim with such precision is not essential. Therefore, we opt not to add this additional figure. We note that the comparison between Figures 16 and 17 is rather simple despite the difference in viewing angle, as the flow features are quite distinctive. Furthermore, readers who wish to examine the exact values and positions in detail can do so using the data provided in the accompanying repository.

12. In Section 3.4, the reviewer urges to know more about the vortex dynamics shed by MRSLs, especially on the mutual induction of the trialing vortices. The reviewer also made some comments on the vortex behaviors based on his or hers observations.

**Reply:**

We thank the reviewer for their interest in the vortex dynamics revealed by our data, and we also appreciate their recognition of the value of our experimental dataset. As the reviewer observed, the three pairs of streamwise vortical structures indeed undergo rotation/precession, with the UW and DW rotate in counter directions. In the original manuscript, we did not explore this aspect in detail, as we considered it less central to the main objective and conclusions of the paper. Nevertheless, we fully acknowledge that the fluid mechanics community has a natural interest in vortex dynamics. Motivated by the reviewer's comments, we have added a new appendix (Appendix D) that presents and discusses the vortex dynamics in greater detail, including the mutual induction and spatial evolution of the trailing vortices.

13. In line 448, the reviewer is wondering about whether configs UW or DW perform better based on the available power measured within the MRSL's frontal projection area.

**Reply:**

We thank the reviewer for raising this very practical question regarding the application of MRSL. At this stage, we cannot provide a definitive answer. For instance, the force measurements in Figures 11 and 12 show that the UW configuration performs better than DW, which also aligns with the available power before x/D = 10 in Figure 18. However, beyond x/D

= 10, the area-averaged available power suggests that DW begins to outperform UW. These contrasting results complicate a straightforward judgment. Moreover, only simple layouts of UW and DW are tested in this study. A thorough performance assessment would require examining a wider range of effective wind farm layouts under varying wind directions, which lies beyond the scope of the present work.

The main goal of this manuscript is to demonstrate the potential of introducing lifting devices into wind power plants, rather than to provide detailed optimization strategies. At this early stage, proving the capability of MRSL and motivating further investment in this concept are far more critical than determining whether UW or DW is superior. For now, it may be fair to conclude that either UW or DW can be chosen, as both yield broadly comparable performance and both significantly outperform WL (the baseline configuration without lifting devices). We have now stated this point in the Conclusions section, while leaving the final answer open for future research.

14. In section 3.6.2, the reviewer shares his concern about whether it is valid when we compares our experimental results with the previous numerical results when there are significant geometry differences. He or she is also not sure if this comparison strengthen the current work.

**Reply:**

We thank the reviewer for sharing this thoughtful concern.

Although there are indeed significant differences between the simulation cases and the present experiments (e.g., Reynolds number, wing number, and geometric details), we believe the comparison remains meaningful. This is because the key parameter governing MRSL performance is thought to be the lift-to-thrust ratio. While this hypothesis has not yet been systematically validated, our tests suggest that it generally holds true, and we believe it merits dedicated future research. The data presented in Figure 19 (Figure 20 in the revised version) may thus be regarded as preliminary evidence, which is one of the major reasons we include this comparison.

Regarding the necessity of this section, we acknowledge that its added value may appear limited if the current article is viewed as a standalone work. However, the overarching objective of this manuscript is to consolidate the potential of MRSL and RGWF. Prior studies on RGWFs rely heavily if not solely on numerical simulations, and experimental validation has been lacking. Including this section therefore allows us to strengthen the credibility of those earlier findings. Moreover, while the present experiments provide valuable insights, they also have limitations (e.g., discontinuities from different measurement sessions), which introduce uncertainty. We consider the strong agreement observed between the experimental data and high-fidelity simulations as supporting evidence that such imperfections do not undermine the main conclusions of this work.

In light of the reviewer's comment, we recognize that the original wording may not have made the objective of this section sufficiently clear. We have therefore rephrased the text to emphasize that the purpose of this comparison is not to claim a perfect match, but rather to reinforce the broader conclusions drawn in our previous numerical works. This is especially done in the newly introduced text in the first paragraph of Section 3.6.2 of the revised version.

15. In line 581, the reviewer suggest to include a chart describing the available power for the staggered case.

**Reply:**

We thank the reviewer for this helpful suggestion. A chart of the available power for the staggered case has now been included (Figure 24 in the revised version). We believe this addition provides deeper quantitative insight into the performance of RGWFs under a staggered layout.

16. In line 610, the reviewer suggest that the detail interaction between the rotor wake of multirotor and lifting devices may also be interested.

**Reply:**

We thank the reviewer for pointing this out. This has now been included in the text.

**Reply to comments from Reviewer #2**

We sincerely thank the reviewer's recognition and his or hers dedication in reviewing our work.

The comments of the reviewer is as following.

1. Section 2.2 - line 67: a typo for multi-rotor system

**Reply:**

We have addressed the typo.

2. Section 2.4 - line 122: the measurement devices are not clearly reported. The strain gauges are 10N range with a precision of 0.1N? Did you checked for a better calibration? Is the frequency of the horizontal strain gauge 1000Hz or 1Hz? is the frequency of the vertical bench scale only 1Hz?

**Reply:**

We thank the reviewer for pointing this out. The accuracy of our strain gauge sensor is indeed better than 0.1 N. According to the datasheet, the precision is 0.01 N. Similarly, after rechecking the datasheet, we found that the precision of the bench scale is 0.001 N rather than 0.0004 N. Correcting these values has negligible, or even positive, effects on consolidating the analyses and conclusions presented.

Regarding the sampling frequency, the horizontal strain gauge operates at 1,000 Hz, as it is fully programmable and automated. For the bench scale, although it is also technically programmable, we did not use that function because the corresponding software requires subscription fee. Instead, the measurements were taken by manually triggering the recording function. We did not mention this in the manuscript as we find it not necessary for the other to replicate our measurement.